# BOME! Bilevel Optimization Made Easy:
# A Simple First-Order Approach

**Bo Liu**[1][*]  **Mao Ye**[1][*]  **Stephen Wright**[2]  **Peter Stone**[1,3]  **Qiang Liu**[1]
[1]The University of Texas at Austin  [2]University of Wisconsin-Madison  [3] Sony AI
[1]{bliu,my21,pstone,lqiang}@cs.utexas.edu,  [2]swright@cs.wisc.edu

## Abstract

Bilevel optimization (BO) is useful for solving a variety of important machine learning problems including but not limited to hyperparameter optimization, meta-learning, continual learning, and reinforcement learning. Conventional BO methods need to differentiate through the low-level optimization process with implicit differentiation, which requires expensive calculations related to the Hessian matrix. There has been a recent quest for first-order methods for BO, but the methods proposed to date tend to be complicated and impractical for large-scale deep learning applications. In this work, we propose a simple first-order BO algorithm that depends only on first-order gradient information, requires no implicit differentiation, and is practical and efficient for large-scale non-convex functions in deep learning. We provide a non-asymptotic convergence analysis of the proposed method to stationary points for non-convex objectives and present empirical results that show its superior practical performance.

## 1 Introduction

We consider the bilevel optimization (BO) problem:

$$\min_{v,\theta} f(v,\theta) \quad s.t. \quad \theta \in \arg\min_{\theta'} g(v,\theta'), \tag{1}$$

where the goal is to minimize an *outer objective* $f$ whose variables include the solution of another minimization problem w.r.t an *inner objective* $g$. The $\theta$ and $v$ are the *inner* and *outer* variables, respectively. We assume that $v \in \mathbb{R}^m, \theta \in \mathbb{R}^n$ and that $g(v,\cdot)$ attains a minimum for each $v$.

BO is useful in a variety of machine learning tasks. A canonical example is hyperparameter optimization, in which case $f$ (resp. $g$) is the validation (resp. training) loss associated with a model parameter $\theta$ and a hyperparameter $v$, and we want to find the optimal hyperparameter $v$ to minimize the validation loss $f$ when $\theta$ is determined by minimizing the training loss; see e.g., Pedregosa [42], Franceschi et al. [12]. Other applications include meta learning [12], continual learning [43], reinforcement learning [52], and adversarial learning [23]. See Liu et al. [32] for a recent survey.

BO is notoriously challenging due to its nested nature. Despite the large literature, most existing methods for BO are slow and unsatisfactory in various ways. For example, a major class of BO methods is based on direct gradient descent on the outer variable $v$ while viewing the optimal inner variable $\theta^*(v) = \arg\min_\theta g(v,\theta)$ as a (uniquely defined) function of $v$. The key difficulty is to calculate the derivative $\nabla_v \theta^*(v)$ which may require expensive manipulation of the Hessian matrix of $g$ via the implicit differentiation theorem. Another approach is to replace the low level optimization with the stationary condition $\nabla_\theta g(v,\theta) = 0$. This still requires Hessian information, and more importantly, is unsuitable for nonconvex $g$ since it allows $\theta$ to be any stationary point of $g(v,\cdot)$,

---

[*]Equal contribution. MY mainly contributes on developing theory and BL mainly contributes on conducting experiment. Both authors contribute equally on paper writing.

36th Conference on Neural Information Processing Systems (NeurIPS 2022).

not necessarily a minimizer. To the best of our knowledge, the only existing fully first-order BO algorithms[2] are BSG-1 [15] and BVFSM with its variants [33–35]; but BSG-1 relies on a non-vanishing approximation that does not yield convergence to the correct solution in general, and BVFSM is sensitive to hyper-parameters on large-scale practical problems and lacks a complete non-asymptotic analysis for the practically implemented algorithm.

In this work, we seek a *simple and fast fully first-order* BO method that can be used with non-convex functions including those appear in deep learning applications. The idea is to reformulate (1) as a single-level constrained optimization problem using the so-called value-function-based approach [10, 7]. The constrained problem is then solved by stopping gradient on the single variable that contains the higher-order information and applying a simple first-order dynamic barrier gradient descent method based on a method of Gong et al. [16]. Our contributions are: **1)** we introduce a novel and fast BO method by applying a modified dynamic barrier gradient descent on the value-function reformulation of BO; **2)** Theoretically, we establish the non-asymptotic convergence of our method to local stationary points (as measured by a special KKT loss) for non-convex $f$ and $g$. Importantly, to the best of our knowledge, this work is the first to establish non-asymptotic convergence rate for a fully first-order BO method. This result is also much beyond that of Gong et al. [16] and Ji et al. [22]. **3)** Empirically, the proposed method achieves better or comparable performance while being more efficient than state-of-the-art BO methods on a variety of benchmarks.

## 2   Background

This section provides a brief background on traditional BO methods. Please see Bard [2], Dempe and Zemkoho [7], Dempe [6] for overviews, and Liu et al. [32] for a survey on recent ML applications.

**Hypergradient Descent** Assume that the minimum of $g(v, \cdot)$ is unique for all $v$ so that we can write $\theta^*(v) = \arg\min_\theta g(v, \theta)$ as a function of $v$; this is known as the low-level singleton (LLS) assumption. The most straightforward approach to solving (1) is to conduct gradient descent on $f(v, \theta^*(v))$ as a function of $v$. Note that

$$\nabla_v f(v, \theta^*(v)) = \nabla_1 f(v, \theta^*(v)) + \nabla_v \theta^*(v) \nabla_2 f(v, \theta^*(v)).$$

The difficulty is to compute $\nabla_v \theta^*(v)$. From implicit function theorem, it satisfies a linear equation:

$$\nabla_{1,2} g(v, \theta^*(v)) + \nabla_{2,2} g(v, \theta^*(v)) \nabla_v \theta^*(v) = 0. \tag{2}$$

If $\nabla_{2,2} g$ is invertible, we can solve for $\nabla_v \theta^*(v)$ and obtain a gradient update rule on $v$:

$$v_{k+1} \leftarrow v_k - \xi \left( \nabla_1 f_k - \left( \nabla_{1,2} g_k \right)^\top \left( \nabla_{2,2} g_k \right)^{-1} \nabla_2 f_k \right),$$

where $k$ denotes iteration, $\nabla_1 f_k = \nabla_1 f(v_k, \theta^*(v_k))$ and similarly for the other terms. This approach is sometimes known as the *hypergradient descent*. However, hypergradient descent is computationally expensive: Besides requiring evaluation of the inner optimum $\theta^*(v_k)$, the main computational bottleneck is to solve the linear equation in (2). Methods have been developed that approximate (2) using conjugate gradient [42, 44, 17], Neumann series [29, 37], and related variants [14]. Another popular approximation approach is to replace $\nabla_v \theta^*(v)$ with $\nabla_v \theta^{(T)}(v)$, where $\theta^{(T)}(v)$ denotes the $T$-th iteration of gradient descent or other optimization steps on $g(v, \theta)$ w.r.t. $\theta$ starting from certain initialization. The gradient $\nabla_v \theta^{(T)}(v)$ can be calculated with auto-differentiation (AD) with either forward mode [11], backward mode [12, 11, 46, 28, 1] or their variants [31]. While these approaches claim to be first-order, they require many Hessian-vector or Jacobian-vector products at each iteration and are slow for large problems.

Other examples of approximation methods include a neural surrogate method which approximates $\theta^*(v)$ and its gradient $\nabla_v \theta^*(v)$ with neural networks [38] and Newton-Gaussian approximation of the Hessian matrix with covariance of gradient [15]. Both approaches introduce non-vanishing approximation error that is difficult to control. The neural surrogate method also suffers from high training cost for the neural network.

**Stationary-Seeking Methods.** An alternative method is to replace the argmin constraint in (1) with the stationarity condition $\nabla_\theta g(v, \theta) = 0$, yielding a constrained optimization:

$$\min_{v, \theta} f(v, \theta) \quad s.t. \quad \nabla_\theta g(v, \theta) = 0. \tag{3}$$

---

[2]By fully first-order, we mean methods that only require information of $f, g, \nabla f, \nabla g$, so this excludes methods that apply auto-differentiation or conjugate gradient that need multiple steps of matrix-vector computation.

Algorithms for nonlinear equality constrained optimization can then be applied [39]. The constraint in (3) guarantees only that $\theta$ is a stationary point of $g(v, \cdot)$, so it is equivalent to (1) only when $g$ is convex w.r.t. $\theta$. Otherwise, the solution of (3) can be a maximum or saddle point of $g$. This makes it problematic for deep learning, where non-convex functions are pervasive.

## 3 Method

We consider a *value function approach* [see e.g., 41, 53, 33], which yields natural first-order algorithms for non-convex $g$ and requires no computation of Hessian matrices. It is based on the observation that (1) is equivalent to the following constrained optimization (even for non-convex $g$):

$$\min_{v,\theta} f(v,\theta) \quad s.t. \quad q(v,\theta) := g(v,\theta) - g^*(v) \leq 0, \tag{4}$$

where $g^*(v) := \min_\theta g(v,\theta) = g(v, \theta^*(v))$ is known as the value function. Compared with the hypergradient approach, this formulation **does not require calculation of the implicit derivative** $\nabla_v \theta^*(v)$: Although $g^*(v)$ depends on $\theta^*(v)$, its derivative $\nabla_v g^*(v)$ does not depend on $\nabla_v \theta^*(v)$, by Danskin's theorem:

$$\nabla_v g^*(v) = \nabla_1 g(v, \theta^*(v)) + \nabla_v \theta^*(v) \nabla_2 g(v, \theta^*(v)) = \nabla_1 g(v, \theta^*(v)), \tag{5}$$

where the second term in (5) vanishes because we have $\nabla_2 g(v, \theta^*(v)) = 0$ by definition of the optimum $\theta^*(v)$. Therefore, provided that we can evaluate $\theta^*(v)$ at each iteration, solving (4) yields an algorithm for BO that requires no Hessian computation. In this work, we make use of the dynamic barrier gradient descent algorithm of Gong et al. [16] to solve (4). This is an elementary first-order algorithm for solving constrained optimization, but it applies only to a special case of the bilevel problem and must be extended to handle the general case we consider here.

**Dynamic Barrier Gradient Descent.** The idea is to iterative update the parameter $(v, \theta)$ to reduce $f$ while controlling the decrease of the constraint $q$, ensuring that $q$ decreases whenever $q > 0$. Specifically, denote $\xi$ as the step size, the update at each step is

$$(v_{k+1}, \theta_{k+1}) \leftarrow (v_k, \theta_k) - \xi \delta_k, \tag{6}$$

$$\text{where} \quad \delta_k = \arg\min_\delta \|\nabla f(v_k, \theta_k) - \delta\|^2 \quad \text{s.t.} \quad \langle \nabla q(v_k, \theta_k), \delta \rangle \geq \phi_k. \tag{7}$$

Here $\nabla f_k := \nabla_{(v,\theta)} f(v_k, \theta_k)$, $\nabla q_k := \nabla_{(v,\theta)} q(v_k, \theta_k)$, and $\phi_k \geq 0$ is a non-negative control barrier and should be strictly positive $\phi_k > 0$ in the non-stationary points of $q$: the lower bound on the inner product of $\nabla q(v_k, \theta_k)$ and $\delta_k$ ensures that the update in (6) can only decrease $q$ (when step size $\xi$ is sufficiently small) until it reaches stationary. In addition, by enforcing $\delta_k$ to be close to $\nabla f(v_k, \theta_k)$ in (7), we decrease the objective $f$ as much as possible so long as it does not conflict with descent of $q$.

Two straightforward choices of $\phi_k$ that satisfies the condition above are $\phi_k = \eta q(v_k, \theta_k)$ and $\phi_k = \eta \|\nabla q(v_k, \theta_k)\|^2$ with $\eta > 0$. We find that both choices of $\phi_k$ work well empirically and use $\phi_k = \eta \|\nabla q(v_k, \theta_k)\|^2$ as the default (see Section 6.2).

The optimization in (7) yields a simple closed form solution:

$$\delta_k = \nabla f(v_k, \theta_k) + \lambda_k \nabla q(v_k, \theta_k), \quad \text{with} \quad \lambda_k = \max\left(\frac{\phi_k - \langle \nabla f(v_k, \theta_k), \nabla q(v_k, \theta_k) \rangle}{\|\nabla q(v_k, \theta_k)\|^2}, \ 0\right),$$

and $\lambda_k = 0$ in the case of $\|\nabla q(v_k, \theta_k)\| = 0$.

**Practical Approximation.** The main bottleneck of the method above is to calculate the $q(v_k, \theta_k)$ and $\nabla q(v_k, \theta_k)$ which requires evaluation of $\theta^*(v_k)$. In practice, we approximate $\theta^*(v_k)$ by $\theta_k^{(T)}$, where $\theta_k^{(T)}$ is obtained by running $T$ steps of gradient descent of $g(v_k, \cdot)$ w.r.t. $\theta$ starting from $\theta_k$. That is, we set $\theta_k^{(0)} = \theta_k$ and let

$$\theta_k^{(t+1)} = \theta_k^{(t)} - \alpha \nabla_\theta g(v_k, \theta_k^{(t)}), \quad t = 0, \ldots, T-1, \tag{8}$$

for some step size parameter $\alpha > 0$. We obtain an estimate of $q(v, \theta)$ at iteration $k$ by replacing $\theta^*(v_k)$ with $\theta_k^{(T)}$: $\hat{q}(v, \theta) = g(v, \theta) - g(v, \theta_k^{(T)})$.

---

**Algorithm 1** Bilevel Optimization Made Easy (BOME!)

---

**Goal**: Solve $\min_{v,\theta} f(v,\theta)$ $s.t.$ $\theta \in \arg \min g(v,\cdot)$.

**Input**: Initialization $(v_0, \theta_0)$; inner step $T$; outer and inner stepsize $\xi$, $\alpha$ (set $\alpha = \xi$ by default).

**for** iteration $k$ **do**

    1. Get $\theta_k^{(T)}$ by $T$ steps of gradient descent on $g(v_k, \cdot)$ starting from $\theta_k$ (See Eq. (8)).

    2. Set $\hat{q}(v,\theta) = g(v,\theta) - g(v, \theta_k^{(T)})$.

    3. Update $(v,\theta)$ : $(v_{k+1}, \theta_{k+1}) \leftarrow (v_k, \theta_k) - \xi(\nabla f(v_k, \theta_k) + \lambda_k \nabla \hat{q}(v_k, \theta_k))$

$$\text{where} \quad \lambda_k = \max\left( \frac{\phi_k - \langle \nabla f(v_k, \theta_k), \ \nabla \hat{q}(v_k, \theta_k) \rangle}{\|\nabla \hat{q}(v_k, \theta_k)\|^2}, \ 0 \right),$$

    and $\phi_k = \eta \|\nabla \hat{q}(v_k, \theta_k)\|^2$ (default), or $\phi_k = \eta \hat{q}(v_k, \theta_k)$ with $\eta > 0$.

    **Remark**: 1) We treat $\theta_k^{(T)}$ as constant when taking derivative of $\hat{q}$; 2) In practice, step 3 can have separate stepsize $(\xi_v, \xi_\theta)$ and use standard optimizers like Adam [26]; 3) We use $\eta = 0.5$ and $T = 10$ by default.

**end for**

---

We substitute $\hat{q}(v_k, \theta_k)$ into (7) to obtain the update direction $\delta_k$. The full procedure is summarized in Algorithm 1. Note that the $\theta_k^{(T)}$ is viewed as a constant when defining $\hat{q}(v,\theta)$ and hence no differentiation of $\theta_k^{(T)}$ is performed when calculating the gradient $\nabla \hat{q}$. This differs from truncated back-propagation methods [e.g.,[46]] which differentiate through $\theta_k^{(T)}$ as a function of $v$. Alternatively, it can be viewed as a plug-in estimator. We know that

$$\begin{aligned}
\nabla_{v_k} q(v_k, \theta_k) &= \nabla_{v_k} g(v_k, \theta_k) - \nabla_{v_k} g(v_k, \theta^*(v_k)) \\
&= \nabla_{v_k} g(v_k, \theta_k) - [\nabla_1 g(v_k, \theta^*(v_k)) + \nabla_{v_k} \theta^*(v_k) \nabla_2 g(v_k, \theta^*(v_k))] \\
&= \nabla_{v_k} g(v_k, \theta_k) - \nabla_1 g(v_k, \theta^*(v_k)),
\end{aligned}$$

where $\nabla_1$ denotes taking the derivative w.r.t. the first variable. Since $\theta^*(v_k)$ is unknown, we estimate $\nabla_1 g(v_k, \theta^*(v_k))$ by plugging-in $\theta_k^{(T)}$ to approximate $\theta^*(v_k)$:

$$\nabla_{v_k} \hat{q}(v_k, \theta_k) = \nabla_{v_k} g(v_k, \theta_k) - \nabla_1 g(v_k, \theta^*(v_k)).$$

Each step of Algorithm 1 can be viewed as taking one step (starting from $v_k, \theta_k$) toward solving an approximate constrained optimization problem:

$$\min_{v,\theta} f(v,\theta) \quad s.t. \quad g(v,\theta) \leq g(v, \theta_k^{(T)}), \tag{9}$$

which can be viewed as a relaxation of the exact constrained optimization formulation (4), because $\{(v,\theta)\colon g(v,\theta) \leq g^*(v)\}$ is a subset of $\{(v,\theta)\colon g(v,\theta) \leq g(v, \theta_k^{(T)})\}$.

## 4 Analysis

We first elaborate the KKT condition of (4) (Section 4.1), then quantify the convergence of the method by how fast it meets the KKT condition. We consider both the case when $g$ satisfies the Polyak-Łojasiewicz (PL) inequality w.r.t. $\theta$, hence having a unique global optimum (Section 4.2), and when $g$ have multiple local minimum (Section 4.3).

### 4.1 KKT Conditions

Consider a general constrained optimization of form $\min f(v,\theta)$ s.t. $q(v,\theta) \leq 0$. Under proper regularity conditions known as constraint quantifications [40], the first-order KKT condition gives a necessary condition for a feasible point $(v^*, \theta^*)$ with $q(v^*, \theta^*) \leq 0$ to be a local optimum of (4): There exists a Lagrangian multiplier $\lambda^* \in [0, +\infty)$, such that

$$\nabla f(v^*, \theta^*) + \lambda^* \nabla q(v^*, \theta^*) = 0, \tag{10}$$

and $\lambda^*$ satisfies the complementary slackness condition $\lambda^* q(v^*, \theta^*) = 0$. A common regularity condition to ensure (10) is the *constant rank constraint quantification (CRCQ)* condition [20].

**Definition 1.** *A point $(v^*, \theta^*)$ is said to satisfy CRCQ with a function $h$ if the rank of the Jacobian matrix $\nabla h(v, \theta)$ is constant in a neighborhood of $(v^*, \theta^*)$.*

Unfortunately, **the KKT condition in** (10) **does not hold for the bilevel optimization in** (4). The CRCQ condition does not typically hold for this problem. This is because the minimum of $q$ is zero, and hence if $(v^*, \theta^*)$ is feasible for (4), then $(v^*, \theta^*)$ must attain the minimum of $q$, yielding $q(v^*, \theta^*) = 0$ and $\nabla q(v^*, \theta^*) = 0$ if $q$ is smooth; but we could not have $\nabla q(v, \theta) = 0$ uniformly in a neighborhood of $(v^*, \theta^*)$ (hence CRCQ fails) unless $q$ is a constant around $(v^*, \theta^*)$. In addition, if KKT (10) holds, we would have $\nabla f(v^*, \theta^*) = -\lambda^* \nabla q(v^*, \theta^*) = 0$ which happens only in the rare case when $(v^*, \theta^*)$ is a stationary point of both $f, g$.

Instead, one can establish a KKT condition of BO through the form in (3), because there is nothing special that prevents $(v^*, \theta^*)$ from satisfying CRCQ with $\nabla_\theta q = \nabla_\theta g$ (even though we just showed that it is difficult to have CRCQ with $q$). Assume $f$ and $\nabla_\theta q$ are continuously differentiable, and $(v^*, \theta^*)$ is a point satisfying $\nabla_\theta q(v^*, \theta^*) = 0$ and CRCQ with $\nabla_\theta q$. Then by the typical first order KKT condition of (3), there exists a Lagrange multiplier $\omega^* \in \mathbb{R}^n$ such that

$$\nabla f(v^*, \theta^*) + \nabla(\nabla_\theta q(v^*, \theta^*))\omega^* = 0. \tag{11}$$

This condition can be viewed as the limit of a sequence of (10) in the following way: assume we relax the constraint in (4) to $q(v, \theta) \leq c_k$ where $c_k$ is a sequence of positive numbers that converge to zero, then we can establish (10) for each $c_k > 0$ and pass the limit to zero to yield (11).

**Proposition 1.** *Assume that $f, q, \nabla q$ are continuously differentiable and $\|\nabla f\|, f$ is bounded. For a feasible point $(v^*, \theta^*)$ of (4) that satisfies CRCQ with $\nabla_\theta q$, if $(v^*, \theta^*)$ is the limit of a sequence $\{(v_k, \theta_k)\}_{k=1}^\infty$ satisfying $q(v_k, \theta_k) \neq 0 \;\forall k$, and there exists a sequence $\{\lambda_k\} \subset [0, \infty)$ such that*

$$\nabla f(v_k, \theta_k) + \lambda_k \nabla q(v_k, \theta_k) \to 0, \qquad\qquad q(v_k, \theta_k) \to 0,$$

*as $k \to +\infty$, then $(v^*, \theta^*)$ satisfies (11).*

This motivates us to use the following function as a measure of stationarity of the solution returned by the algorithm:

$$\mathcal{K}(v, \theta) = \underbrace{\min_{\lambda \geq 0} \|\nabla f(v, \theta) + \lambda \nabla q(v, \theta)\|^2}_{\text{local improvement}} + \underbrace{q(v, \theta)}_{\text{feasibility}}.$$

The hope is to have an algorithm that generates a sequence $\{(v_k, \theta_k)\}_{k=0}^\infty$ that satisfies $\mathcal{K}(v_k, \theta_k) \to 0$ as $k \to +\infty$.

Intuitively, the first term in $\mathcal{K}(v, \theta)$ measures how much $\nabla f$ conflicts with $\nabla q$ (how much we can decrease $f$ without increasing $q$), as it is equal to the squared $\ell_2$ norm of the solution to the problem $\min_\delta \|\nabla f - \delta\|^2$ s.t. $\langle \nabla q, \delta \rangle \geq 0$. The second term in $\mathcal{K}$ measures how much the $\arg\min g$ constraint is satisfied.

## 4.2 Convergence with unimodal $g$

We first present the convergence rate when assuming $g(v, \cdot)$ has unique minimizer and satisfies the Polyak-Łojasiewicz (PL) inequality for all $v$, which guarantees a linear convergence rate of the gradient descent on the low level problem.

**Assumption 1** (PL-inequality). *Given any $v$, assume $g(v, \cdot)$ has a unique minimizer denoted as $\theta^*(v)$. Also assume there exists $\kappa > 0$ such that for any $(v, \theta)$, $\|\nabla_\theta g(v, \theta)\|^2 \geq \kappa(g(v, \theta) - g(v, \theta^*(v)))$.*

The PL inequality gives a characterization on how a small gradient norm implies global optimality. It is implied from, but weaker than strongly convexity. The PL-inequality is more appealing than convexity because some modern over-parameterized deep neural networks have been shown to satisfy the PL-inequality along the trajectory of gradient descent. See, for example, Frei and Gu [13], Song et al. [48], Liu et al. [30] for more discussion.

**Assumption 2** (Smoothness). *$f$ and $g$ are differentiable, and $\nabla f$ and $\nabla g$ are $L$-Lipschitz w.r.t. the joint inputs $(v, \theta)$ for some $L \in (0, +\infty)$.*

**Assumption 3** (Boundedness). *There exists a constant $M < \infty$ such that $\|\nabla g(v, \theta)\|$, $\|\nabla f(v, \theta)\|$, $|f(v, \theta)|$ and $|g(v, \theta)|$ are all upper bounded by $M$ for any $(v, \theta)$.*

Assumptions 2 and 3 are both standard in optimization.

**Theorem 1.** *Consider Algorithm 1 with $\xi, \alpha \leq 1/L$, $\phi_k = \eta \|\nabla \hat{q}(v_k, \theta_k)\|^2$, and $\eta > 0$. Suppose that Assumptions 1, 2, and 3 hold. Then there exists a constant $c$ depending on $\alpha, \kappa, \eta, L$ such that when $T \geq c$, we have for any $K \geq 0$,*

$$\min_{k \leq K} \mathcal{K}(v_k, \theta_k) = O\left( \sqrt{\xi} + \sqrt{\frac{q_0}{\xi K}} + \frac{1}{\xi K} + \exp(-bT) \right)$$

*where $q_0 = q(v_0, \theta_0)$, and $b > 0$ is a constant depending on $\kappa$, $L$, and $\alpha$.*

**Remark** Note that one of the dominant terms depends on the initial value $q_0 = q(v_0, \theta_0)$. Therefore, we can obtain a better rate if we start from a $\theta_0$ with small $q_0$ (hence near the optimum of $g(v_0, \cdot)$). In particular, when $q(v_0, \theta_0) = O(1)$, choosing $\xi = O(K^{-1/2})$ gives $\min_{k \leq K} \mathcal{K}(v_k, \theta_k) = O(K^{-1/4} + \exp(-bT))$ rate. On the other hand, if we start from a better initialization such that $q(v_0, \theta_0) = O((\xi K)^{-1})$, then choosing $\xi = O(K^{-2/3})$ gives $\min_{k \leq K} \mathcal{K}(v_k, \theta_k) = O(K^{-1/3} + \exp(-bT))$.

### 4.3  Convergence with multimodal $g$

The PL-inequality eliminates the possibility of having stationary points that are not global optimum. To study cases in which $g$ has multiple local optima, we introduce the notion of attraction points following gradient descent.

**Definition 2** (Attraction points). *Given any $(v, \theta)$, we say that $\theta^\diamond(v, \theta)$ is the attraction point of $(v, \theta)$ with step size $\alpha > 0$ if the sequence $\{\theta^{(t)}\}_{t=0}^\infty$ generated by gradient descent $\theta^{(t)} = \theta^{(t-1)} - \alpha \nabla_\theta g(v, \theta^{(t-1)})$ starting from $\theta^{(0)} = \theta$ converges to $\theta^\diamond(v, \theta)$.*

Assume the step size $\alpha \leq 1/L$ where $L$ is the smoothness constant defined in Assumption 2, one can show the existence and uniqueness of attraction point of any $(v, \theta)$ using Proposition 1.1 of Traonmilin and Aujol [49]. Intuitively, the attraction of $(v, \theta)$ is where the gradient descent algorithm can not make improvement. In fact, when $\alpha \leq 1/L$, one can show that $g(v, \theta) \leq g(v, \theta^\diamond(v, \theta))$ is equivalent to the stationary condition $\nabla_\theta g(v, \theta) = 0$.

The set of $(v, \theta)$ that have the same attraction point forms an attraction basin. Our analysis needs to assume the PL-inequality within the individual attraction basins.

**Assumption 4** (Local PL-inequality within attraction basins). *Assume that for any $(v, \theta)$, $\theta^\diamond(v, \theta)$ exists. Also assume that there exists $\kappa > 0$ such that for any $(v, \theta)$ $\|\nabla_\theta g(v, \theta)\|^2 \geq \kappa(g(v, \theta) - g(v, \theta^\diamond(v, \theta)))$.*

We can also define local variants of $q$ and $\mathcal{K}$ as follows:

$$q^\diamond(v, \theta) = g(v, \theta) - g(v, \theta^\diamond(v, \theta)), \quad \mathcal{K}^\diamond(v, \theta) = \min_{\lambda \geq 0} \|\nabla f(v, \theta) + \lambda \nabla q^\diamond(v, \theta)\|^2 + q^\diamond(v, \theta).$$

Compared with Section 4.2, a key technical challenge is that $\theta^\diamond(v, \theta)$ and hence $q^\diamond(v, \theta)$ can be discontinuous w.r.t. $\theta$ when it is on the boundary of different attraction basins; $\mathcal{K}^\diamond$ is not well defined on these points. However, these boundary points are not stable stationary points, and it is possible to use arguments based on the stable manifold theorem to show that an algorithm with random initialization will almost surely not visit them [47, 27].

**Theorem 2.** *Consider Algorithm 1 with $\xi, \alpha \leq 1/L$, $\phi_k = \eta \|\nabla \hat{q}(v_k, \theta_k)\|^2$, and $\eta > 0$. Suppose that Assumptions 2, 3, and 4 hold and that $q^\diamond$ is differentiable on $(v_k, \theta_k)$ at every iteration $k \geq 0$. Then there exists a constant $c$ depending on $\alpha, \kappa, \eta, L$, such that when $T \geq c$, we have*

$$\min_{k \leq K} \mathcal{K}^\diamond(v_k, \theta_k) = O\left( \sqrt{\xi} + \sqrt{\frac{1}{\xi K}} + \exp(-bT) \right),$$

*where $b$ is a positive constant depending on $\kappa$, $L$, and $\alpha$.*

Unlike Theorem 1, the rate does not improve when $q_0^\diamond := q^\diamond(v_0, \theta_0)$ is small because the attraction basin may change in different iterations, eliminating the benefit of starting from a good initialization. Choosing $\xi = O(K^{-1/2})$ gives $O(K^{-1/4} + \exp(-bT))$ rate of $\min_{k \leq K} \mathcal{K}^\diamond(v_k, \theta_k)$.

# 5 Related Works

The value-function formulation (4) is a classical approach in bilevel optimization [41, 53, 7]. However, despite its attractive properties, it has been mostly used as a theoretical tool, and much less exploited for practical algorithms compared with the more widely known hypergradient approach (Section 2), especially for challenging nonconvex functions $f$ and $g$ such as those encountered in deep learning. One exception is Liu et al. [33], which proposes a BO method by solving the value-function formulation using an interior-point method combined with a smoothed approximation. This was improved later in a pessimistic trajectory truncation approach [35] and a sequential minimization approach [34] (BVFSM). Similar to our approach, these methods do not require computation of Hessians, thanks to the use of value function. However, as we observe in experiments (Section 6.2), BVFSM tends to be dominated by our method both in accuracy and speed, and is sensitive to some hyperparameters that are difficult to tune (such as the coefficients of the log-barrier function in interior point method). Theoretically, Liu et al. [33, 34, 35] provide only asymptotic analysis on the convergence of the smoothed and penalized surrogate loss to the target loss. They do not give an analysis for the algorithm that was actually implemented.

Our algorithm is build up on the dynamic control barrier method of Gong et al. [16], an elementary approach for constrained optimization. Gong et al. [16] also applied their approach to solve a lexicographical optimization of form $\min_\theta f(\theta)$ s.t. $\theta \in \arg\min_{\theta'} g(\theta')$, which is a bilevel optimization without an outer variable (known as *simple bilevel optimization* [8]). Our method is an extension of their method to general bilevel optimization. Such extension is not straightforward, especially when the lower level problem is non-convex, requiring introducing the stop-gradient operation in a mathematically correct way. We also provide non-asymptotic analysis for our method, that goes beyond the continuous time analysis in Gong et al. [16]. A key sophistication in the theoretical analysis is that we need to control the approximation error of $\theta^*(v_k)$ with $\theta_k^{(T)}$ at each step, which requires an analysis significantly different from that of Gong et al. [16]. Indeed, non-asymptotic results have not yet been obtained for many BO algorithms. Even for the classic hypergradient-based approach (such results are established only recently in Ji et al. [22]). We believe that we are the first to establish a non-asymptotic rate for a purely first-order BO algorithm under general assumptions, e.g. the lower level problem can be both convex or non-convex.

Another recent body of theoretical works on BO focus on how to optimize when only stochastic approximation of the objectives is provided [14, 19, 22, 51, 18, 4, 25]; there are also recent works on the lower bounds and minimax optimal algorithms [21, 22]. These algorithms and analysis are based on hypergradient descent and hence require Hessian-vector products in implementation.

# 6 Experiment

We conduct experiments (1) to study the correctness, basic properties, and robustness to hyperparameters of BOME, and (2) to test its performance and computational efficiency on challenging ML applications, compared with state-of-the-art bilevel algorithms. In the following, we first list the baseline methods and how we set the hyperparameters. Then we introduce the experiment problems in Section 6.1, which includes 3 toy problems and 3 ML applications, and provide the experiment results. Finally we summarize observations and findings in Section 6.2.

**Baselines** A comprehensive set of state-of-the-art BO methods are chosen as baseline methods. This includes the *fully first-order* methods: BSG-1 [15] and BVFSM [34], ; a *stationary-seeking* method: Penalty [39], *explicit/implicit* methods: ITD [22], AID-CG (using conjugate gradient), AID-FP (using fixed point method) [17], reverse (using reverse auto-differentiation) [11] stocBiO [22], and VRBO [51].

**Hyperparameters** Unless otherwise specified, BOME strictly follows Algorithm 1 with $\phi_k = \eta \|\nabla \hat{q}(v_k, \theta_k)\|^2$, $\eta = 0.5$, and $T = 10$. The inner stepsize $\alpha$ is set to be the same as outer stepsize $\xi$. The stepsizes of all methods are set by a grid search from the set $\{0.01, 0.05, 0.1, 0.5, 1, 5, 10, 50, 100, 500, 1000\}$. All toy problems adopt vanilla gradient descent (GD) and applications on hyperparameter optimization adapts GD with a momentum of 0.9. Details are provided in Appendix A.

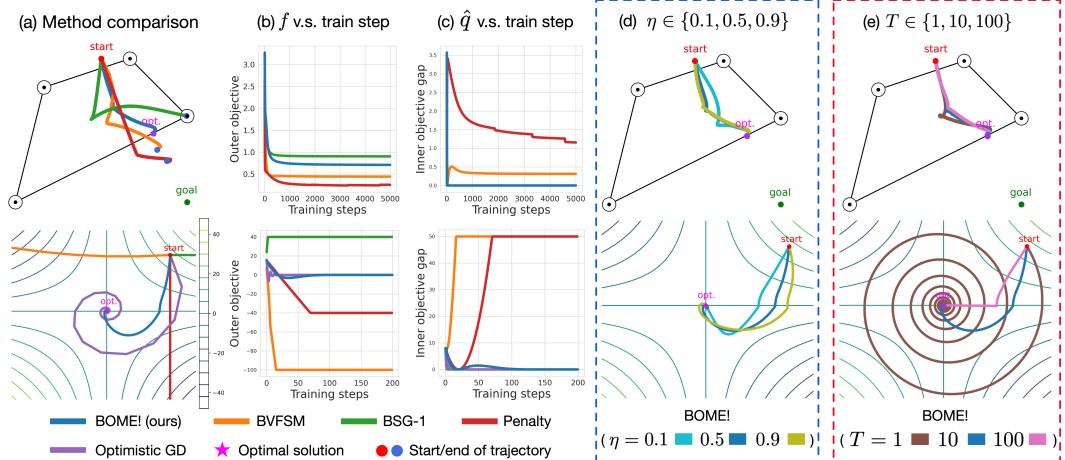

Figure 1: Results on the toy coreset problem and mini-max problem. (a)-(c): the trajectories of $(v_k, \theta_k)$, and $f(v_k, v_k)$ and $\hat{q}_k(v_k, v_k)$ of BOME (our method), BSG-1 [15], BVFSM [34], Penalty [39] and Optimistic GD [5] (only for minimax problem). (d)-(e) trajectories of BOME with different choices of inner gradient step $T$ and the control coefficient $\eta$.

## 6.1 Experiment Problems and Results

**Toy Coreset Problem** To validate the *convergence* property of BOME, we consider:

$$\min_{v,\theta} \|\theta - x_0\|^2 \quad s.t. \quad \theta \in \arg\min_{\theta'} \|\theta' - X\sigma(v)\|^2,$$

where $\sigma(v) = \exp(v)/\sum_{i=1}^4 \exp(v_i)$ is the softmax function, $v \in \mathbb{R}^4, \theta \in \mathbb{R}^2$, and $X = [x_1, x_2, x_3, x_4] \in \mathbb{R}^{2\times 4}$. The goal is to find the closest point to a target point $x_0$ within the convex hull of $\{x_1, \ldots, x_4\}$. See Fig. 6.1 (upper row) for the illustration and results.

**Toy Mini-Max Game** Mini-max game is a special and challenging case of BO where $f$ and $g$ contradicts with each other completely (e.g., $f = -g$). We consider

$$\min_{v,\theta \in \mathbb{R}} v\theta \quad s.t. \quad \theta \in \arg\max_{\theta' \in \mathbb{R}} v\theta'. \tag{12}$$

The optimal solution is $v^* = \theta^* = 0$. Note that the naive gradient descent ascent algorithm diverges to infinity on this problem, and a standard alternative is to use optimistic gradient descent [5]. Figure 6.1 (lower row) shows that BOME works on this problem while other first-order BO methods fail.

**Degenerate Low Level Problem** Many existing BO algorithms require the low level singleton (LLS) assumption, which BOME does not require. To test this, we consider an example from Liu et al. [31]:

$$\min_{v\in\mathbb{R}, \theta\in\mathbb{R}^2} \|\theta - [v; 1]\|_2^2 \quad s.t. \quad \theta \in \arg\min_{(\theta_1', \theta_2')\in\mathbb{R}^2} (\theta_1' - v)^2,$$

where $\theta = (\theta_1, \theta_2)$ and the solution is $v^* = 1, \theta^* = (1, 1)$. See Fig. 4 in Appendix A.3 for the result.

**Data Hyper-cleaning** We are given a noisy training set $\mathcal{D}_{\text{train}} := \{x_i, y_i\}_{i=1}^m$ and a clean validation set $\mathcal{D}_{\text{val}}$. The goal is to optimally weight the training data points so that the model trained on the weighted training set yields good performance on the validation set:

$$\min_{v,\theta} \ell^{\text{val}}(\theta), \quad s.t. \quad \theta = \arg\min_{\theta'} \left\{ \ell^{\text{train}}(\theta', v) + c\|\theta'\|^2 \right\},$$

where $\ell^{\text{val}}$ is the validation loss on $\mathcal{D}_{\text{val}}$, and $\ell^{\text{train}}$ is a weighted training loss: $\ell^{\text{train}} = \sum_{i=1}^m \sigma(v_i)\ell(x_i, y_i, \theta)$ with $\sigma(v) = \text{Clip}(v, [0, 1])$ and $v \in \mathbb{R}^m$. We set $c = 0.001$. For the dataset, we use MNIST [9] (FashionMNIST [50]). We corrupt 50% of the training points by assigning them randomly sampled labels. See Fig. 2 (upper panel) for the results. (Results for FashionMNIST are reported in Appendix A.4.)

**Learnable Regularization** We apply bilevel optimization to learn the optimal regularization coefficient on the twenty newsgroup dataset:[3]

$$\min_{v,\theta} \ell^{\text{val}}(\theta) \quad s.t. \quad \theta \in \arg\min_{\theta'} \left\{ \ell^{\text{train}}(\theta') + \|W_v\theta'\|_2^2 \right\},$$

---

[3]Dataset from https://scikit-learn.org/0.19/datasets/twenty_newsgroups.html.

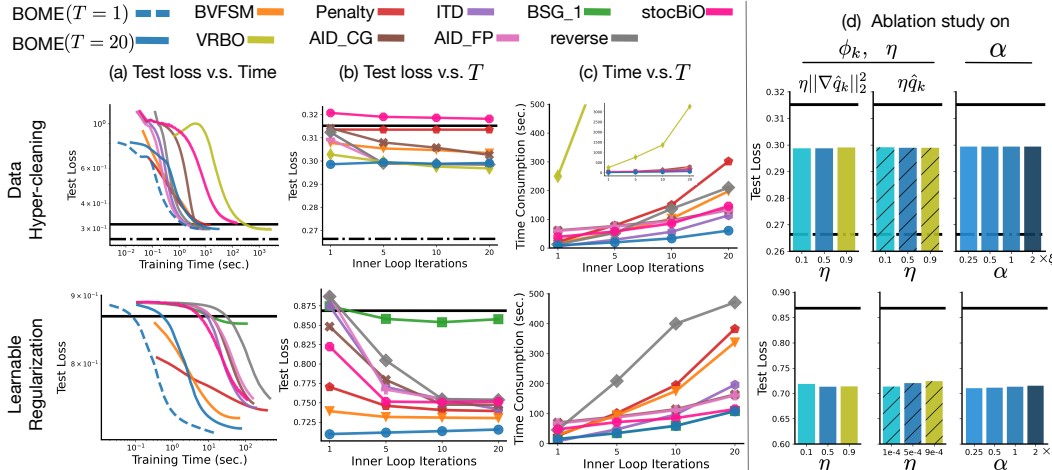

Figure 2: Result for hyperparameter optimization. **Top:** data hyper-cleaning on MNIST dataset. The solid black line is the model performance trained purely on the validation set and the dashed black line is the model performance trained on the validation set and on the part of training set that have correct labels. **Bottom:** learnable regularization on 20 Newsgroup dataset. The solid black line indicates the model performance without any regularization. All results are averaged on 5 random trials. See Appendix A.4 and A.5 for results on FashionMNIST and more details.

| Method | PMNIST | | | Split CIFAR | | |
|---|---|---|---|---|---|---|
| | ACC ($\uparrow$) | NBT ($\downarrow$) | FT ($\uparrow$) | ACC ($\uparrow$) | NBT ($\downarrow$) | FT ($\uparrow$) |
| Offline | $84.95 \pm 0.95$ | - | - | $74.11 \pm 0.66$ | - | - |
| MER | $76.59 \pm 0.74$ | $5.73 \pm 0.59$ | $82.32 \pm 0.34$ | $60.32 \pm 0.86$ | $8.91 \pm 0.86$ | $69.23 \pm 0.40$ |
| CTN (+ITD) | $78.40 \pm 0.28$ | $5.62 \pm 0.39$ | $84.02 \pm 0.29$ | $67.7 \pm 60.96$ | $4.88 \pm 0.77$ | $72.58 \pm 0.62$ |
| CTN (+BVFSM) | $77.78 \pm 0.32$ | $7.25 \pm 0.28$ | $\textbf{85.03} \pm 0.28$ | $67.04 \pm 0.76$ | $6.97 \pm 0.62$ | $\textbf{74.01} \pm 0.57$ |
| CTN (+BOME) | $\textbf{80.70} \pm 0.26$ | $\textbf{4.09} \pm 0.27$ | $84.79 \pm 0.25$ | $\textbf{68.16} \pm 0.60$ | $\textbf{4.72} \pm 0.75$ | $72.88 \pm 0.48$ |

Table 1: Results of continual learning as bilevel optimization. We compute the mean and standard error of each method's results over 5 independent runs. Best results are **bolded**. The full result with comparison against other methods are provided in Table 2 in the Appendix.

where $W_v$ is a matrix depending on $v$, e.g., $W_v = \text{diag}(\exp(v))$. See Fig. 2 (lower panel) for results.

**Continual Learning (CL)** CL studies how to learn on a sequence of tasks in an online fashion without catastrophic forgetting of previously learned tasks. We follow the setting of contextual transformation network (CTN) from Pham et al. [43], which trains a deep neural network consisting of a quickly updated backbone network (parameterized by $\theta$) and a slowly updated controller network (parameterized by $v$). When training the $\tau$-th task, we update $(v, \theta)$ by

$$\min_{v,\theta} \ell^{\text{val}}_{1:\tau}(v, \theta) \quad s.t. \quad \theta \in \arg\min_{\theta'} \ell^{\text{train}}_{1:\tau}(v, \theta'),$$

where $\ell^{\text{train}}_{1:\tau}$ and $\ell^{\text{val}}_{1:\tau}$ are the training and validation loss available up to task $\tau$. The goal is to update the controller such that the long term loss $\ell^{\text{val}}_{1:\tau}$ is minimized assuming $\theta$ is adapted to the available training loss when new tasks come. Assume the CL process terminates at time $t$. Denote by $a^s_\tau$ the test accuracy of task $s$ after training on task $\tau$. We measure the performance of CL by 1) the final mean accuracy on all seen tasks (ACC $= \frac{1}{t} \sum_{\tau \leq t} a^\tau_t$), 2) how much the model forgets as measured by negative backward transfer NBT $= \frac{1}{t} \sum_{\tau \leq t} (a^\tau_\tau - a^\tau_t)$, and 3) how fast the model learns on new tasks as measured by forward transfer FT $= \frac{1}{t} \sum_{\tau \leq t} a^\tau_\tau$. Note that FT = ACC + NBT.

We follow the setting of Pham et al. [43] closely, except replacing their bilevel optimizer (which is essentially ITD [22]) with BOME. See Appendix A.6 for experiment details. The results are shown in Table 1, where in addition to the bilevel algorithms, we also compare with a set of state-of-the-art CL algorithms, including MER [45], ER [3], and GEM [36]. Table 1 also includes an 'Offline" basline – learning $t$ tasks simultaneously using a single model (which is the upper bound on performance).

## 6.2 Observations

**BOME yields faster learning and better solutions at convergence** Figure 6.1-4 show that BOME converges to the optimum of the corresponding bilevel problems and work well on the mini-max optimization and the degenerate low level problem; see also Fig.3 in Appendix A.1. In comparison, the other methods like BSG-1, BVFSM, and Penalty fail to converge to the true optimum even with a grid search over their hyperparameters. Moreover, in all three toy examples, BOME guarantees that $\hat{q}$, which is a proxy for the optimality of the inner problem, decreases to 0. From Fig. 2, it is observed that BOME achieves comparable or better performance than the state-of-the-art bilevel methods for hyperparameter optimization. Moreover, BOME exhibits better computational efficiency (Fig. 2), especially on the twenty newsgroup dataset where the dimension of $\theta$ is large. In Table 1, we find that directly plugging in BOME to the CL problem yields a substantial performance boost.

**Robustness to parameter choices** Besides the standard step size $\xi$ in typical optimizers, BOME only has three parameters: control coefficient $\eta$, inner loop iteration $T$, and inner step size $\alpha$. We use the default setting of $\eta = 0.5$, $T = 10$ and $\alpha = \xi$ across the experiments. From Fig. 6.1 (d,e) and Fig. 2 (b,d), BOME is robust to the choice of $\eta$, $T$ and $\alpha$ as varying them results in almost identical performance. Specifically, $T = 1$ works well in many cases (see Figure 6.1 (e) and 2 (b)). The fact that BOME works well with a small $T$ empirically makes it computationally attractive in practice.

**Choice of control barrier** $\phi_k$ The control barrier is set as $\phi_k = \eta \|\nabla \hat{q}(v_k, \theta_k)\|^2$ by default. Another option is to use $\phi_k = \eta \hat{q}(v_k, \theta_k)$. We test both options on the data hyper-cleaning and learnable regularization experiments in Fig. 2 (d), and observe no significant difference (we choose $\eta$ properly so that both choices of $\phi_k$ is on the same order). Hence we use $\phi_k = \eta \|\nabla \hat{q}(v_k, \theta_k)\|^2$ as the default.

**Comparison against BVFSM** The most relevant baseline to BOME is BVFSM, which similarly adopts the value-function reformulation of the bilevel problems. However, BOME consistently outperform BVFSM in both converged results and computational efficiency, across all experiments. More importantly, BOME has fewer hyperparameters and is robust to them, while we found BVFSM is sensitive to hyperparameters. This makes BOME a better fit for large practical bilevel problems.

## 7 Conclusion and Future Work

BOME, a simple fully first-order bilevel method, is proposed in this work with non-asymptotic convergence guarantee. While the current theory requires the inner loop iterations to scale in a logarithmic order w.r.t to the outer loop iterations, we do not observe this empirically. A further study to understand the mechanism is an interesting future direction.

## 8 Acknowledgement

The research was conducted in both the statistical learning and AI group (SLAI) led by Qiang Liu and the Learning Agents Research Group (LARG) led by Peter Stone in computer science at UT Austin. SLAI research is supported in part by CAREER-1846421, SenSE-2037267, EAGER-2041327, and Office of Navy Research, and NSF AI Institute for Foundations of Machine Learning (IFML). LARG research is supported in part by NSF (CPS-1739964, IIS-1724157, FAIN-2019844), ONR (N00014-18-2243), ARO (W911NF-19-2-0333), DARPA, GM, Bosch, and UT Austin's Good Systems grand challenge. Peter Stone serves as the Executive Director of Sony AI America and receives financial compensation for this work. The terms of this arrangement have been reviewed and approved by the University of Texas at Austin in accordance with its policy on objectivity in research.

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

## Societal Impacts

This paper proposes a simple first order algorithm for bi-level optimization. Many specific instantiation of bi-level optimization such as adversarial learning and data attacking might be harmful to machine learning system in real world, as a general optimization algorithm for bi-level optimization, our method can be a tool in such process. We also develop a great amount of theoretical works and to our best knowledge, we do not observe any significant negative societal impact of our theoretical result.

