# A  Experiment Details

We provide details about each experiment in this section. Regarding the implementation of baseline methods:

- BVFSM's implementation is adapted from `https://github.com/vis-opt-group/BVFSM`.

- Penalty's implementation is adapted from `https://github.com/jihunhamm/bilevel-penalty`.

- VRBO's implementation is adapted from `https://github.com/JunjieYang97/MRVRBO`.

- AID-CG and AID-FP implementations are adapted from `https://github.com/prolearner/hypertorch`.

- ITD implementation is adapted from `https://github.com/JunjieYang97/stocBiO`.

## A.1  Toy Coreset Problem

The problem is:

$$\min_{v,\theta} \|\theta - x_0\|^2 \quad s.t. \quad \theta \in \arg\min_{\theta} \|\theta - X\sigma(v)\|^2,$$

where $\sigma(v) = \exp(v)/\sum_{i=1}^{4} \exp(v_i)$ is the softmax function, $v \in \mathbb{R}^4, \theta \in \mathbb{R}^2$, and $X = [x_1, x_2, x_3, x_4] \in \mathbb{R}^{2\times4}$. where $\sigma(v) = \exp(v)/\sum_{i=1}^{4} \exp(v_i)$ is the softmax function. Here the outer objective $f$ pushes $\theta$ to towards $x_0$ while the inner objective $g$ ensures $\theta$ remains in the convex hull formed by 4 points in the 2D plane (e.g. $X = [x_1, x_2, x_3, x_4] \in \mathbb{R}^{2\times4}$). We choose $x_0 = (3, -2)$ and the four points $x_1 = (1, 3), x_2 = (3, 1), x_3 = (-2, 2)$ and $x_4 = (-3, 2)$. We set $v_0 = (0, 0, 0, 0)$ and $\theta_0 \in \{(0, 3), (-3, 1), (3.5, 1)\}$. For all methods, we fix both the inner stepsize $\alpha$ and the outer stepsize $\xi$ to be 0.05 and set $T = 10$. For BVFSM and Penalty, we grid search the best hyperparameters from $\{0.001, 0.01, 0.1\}$. For BOME, we choose $\phi = \eta \|\nabla\hat{q}\|^2$ and ablate over $\eta \in \{0.1, 0.5, 0.9\}$ and $T \in \{1, 10, 100\}$. The visualization of the optimization trajectories over the 3 initial points are plotted in Fig. 3.

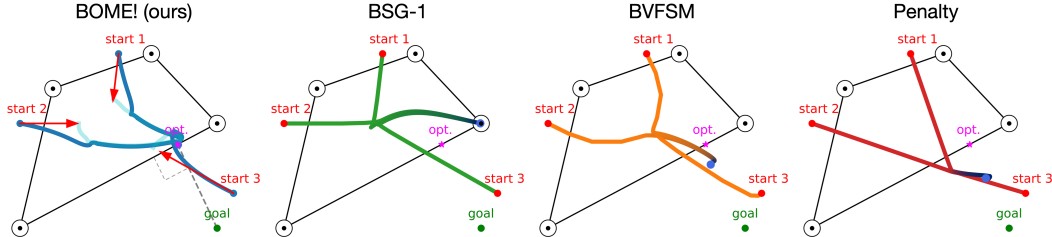

Figure 3: Trajectories of $(v_k, \theta_k)$ on the toy coreset problem (6.1) obtained from BOME (blue) and three recent first-order bilevel methods: BSG-1 [15] (green), BVFSM [34] (orange), and Penalty [39] (red). The goal of the problem is to find the closet point (marked by opt.) to the goal $x_0$ within the convex envelop of the four vertexes. All methods start from 3 initial points (start 1-3), and the converged points are shown in darkblue. For BOME, we also plot the trajectory of $\{\hat{\theta}_k^T\}$ in cyan.

As shown, BOME successfully converges to the optimal solution regardless of the initial $\theta_0$, while BSG-1, BVFSM and Penalty methods converge to non-optimal points. We emphasize that for BVFSM and Penalty, the convergence point *depends on* the choice of hyperparameters.

## A.2  Toy Mini-max Game

The toy mini-max game we consider is:

$$\min_{v\in\mathbb{R}} v\theta^*(v) \quad s.t. \quad \theta^*(v) = \arg\max_{\theta\in\mathbb{R}} v\theta. \tag{13}$$

For BOME and BSG-1, BVFSM, and Penalty methods, we again set both the inner stepsize $\alpha$ and $\beta$ to be 0.05, as no significant difference is observed by varying the stepsizes. For all methods, we set the inner iteration $T = 10$. For BVFSM and Penalty, we grid search the best hyperparameters from $\{0.001, 0.01, 0.1\}$.

## A.3  Without LLS assumption

The toy example to validate whether BOME requires the low-level singleton assumption is borrowed from Liu et al. [31]:

$$\min_{v \in \mathbb{R}, \theta \in \mathbb{R}^2} \|\theta - [v; 1]\|_2^2 \quad s.t. \quad \theta \in \arg\min_{(\theta_1', \theta_2') \in \mathbb{R}^2} (\theta_1' - v)^2,$$

where $\theta = (\theta_1, \theta_2)$ and the optimal solution is $v^* = 1, \theta^* = (1, 1)$. Note that the inner objective has infinite many optimal solution $\theta^*(v)$ since it is degenerated. We set both the inner and outer stepsizes to 0.5 and $T = 10$ for all methods. For BVFSM and Penalty, we grid search the best hyperparameters from $\{0.001, 0.01, 0.1\}$. In Fig. 4, we provide the distance of $f(v_k, \theta_k), g(v_k, \theta_k), \theta_k, v_k$ to their corresponding optimal over training time in seconds. Note that BOME ensures that $\hat{q}(v_k, \theta_k) = g(v_k, \theta_k) - g(v^*, \theta^*)$ decreases to 0.

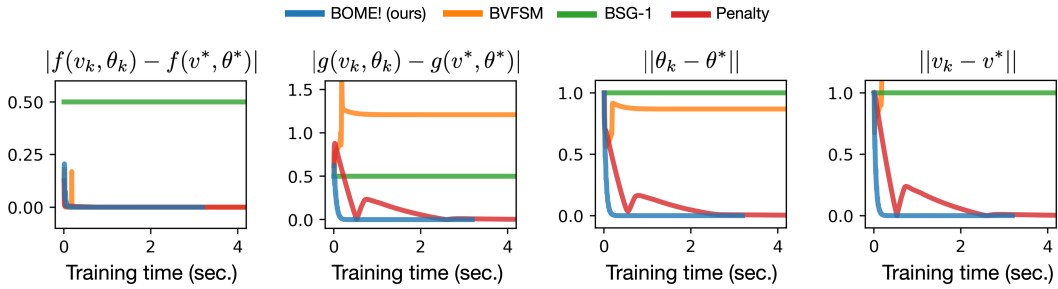

Figure 4: Results on Problem (A.3) which violates the low-level singleton (LLS). We compare BOME against BSG-1, BVFSM, and Penalty. $(v^*, \theta^*)$ denotes the true optimum and The four plots show how fast $f(v_k, \theta_k), g(v_k, \theta_k), \theta_k$ and $v_k$ to the corresponding optimal values w.r.t. the training time in seconds.

## A.4  Data Hyper-cleaning

The bilevel problem for data hyper-cleaning is

$$\min_{v,\theta} \ell^{\text{val}}(\theta), \quad s.t. \quad \theta = \arg\min_\theta \ell^{\text{train}}(\theta, v) + c \|\theta\|^2,$$

where $\ell^{\text{val}}$ is the validation loss on $\mathcal{D}^{\text{val}}$, and $\ell^{\text{train}}$ is a weighted training loss: $\ell^{\text{train}} = \sum_{i=1}^m \sigma(v_i) \ell(x_i, y_i, \theta)$ with $\sigma(v) = \text{Clip}(v, [0, 1])$ and $v \in \mathbb{R}^m$. The training data is of size $m = 50000$ and hence $w \in [0, 1]^{50000}$. The validation data is of size $m = 5000$. The model $\theta = (W, b)$ is a linear model with weight $W$ and bias $b$. Where $W \in \mathbb{R}^{10 \times 784}$ and $b \in \mathbb{R}^{10}$. For this problem, we set inner stepsize $\alpha = 0.01$ for both MNIST and FashionMNIST dataset for all methods as larger or smaller $\alpha$ result in worse performance. As we observe $v$'s gradient norm is much smaller than $\theta$'s in practice, we conduct a grid search over $\xi_v$ from $\{10.0, 50.0, 100.0, 500.0, 1000.0\}$ and also search whether to apply momentum for gradient descent, for all methods. The momentum is searched from $\{0.0, 0.9\}$. For BVFSM and Penalty methods, we also search for their best hyperparameters from $\{0.001, 0.01, 0.1, 1\}$. The model's initial parameter $\theta_0$ is initialized from a pretrained model learned only on the corrupted data. We split the dataset into 4 parts: train set, validation set 1, validation set 2, and the test set. For each method, the model is learned on the train set, and the hyperparameter $v$ is tuned using validation set 1. The best hyperparameter of any algorithm (e.g. stepsize, barrier coefficient, etc.) are then chosen based on the best validation performance on validation set 2. Then we report the final performance of the model on the test set. To conduct the ablation on $\alpha$ for BOME, we search for $\alpha \in \{0.25\xi, 0.5\xi, \xi, 2\xi\}$, where $\xi = 0.01$ is the best stepsize we found for BOME. Results on MNIST and FashionMNIST dataset are provided in Fig. 5. In the first column of Fig. 5, we compare BOME with ($T = 1$ and $T = 20$) with baseline methods whose $T$ is chosen based on best performance on the validation set 2.

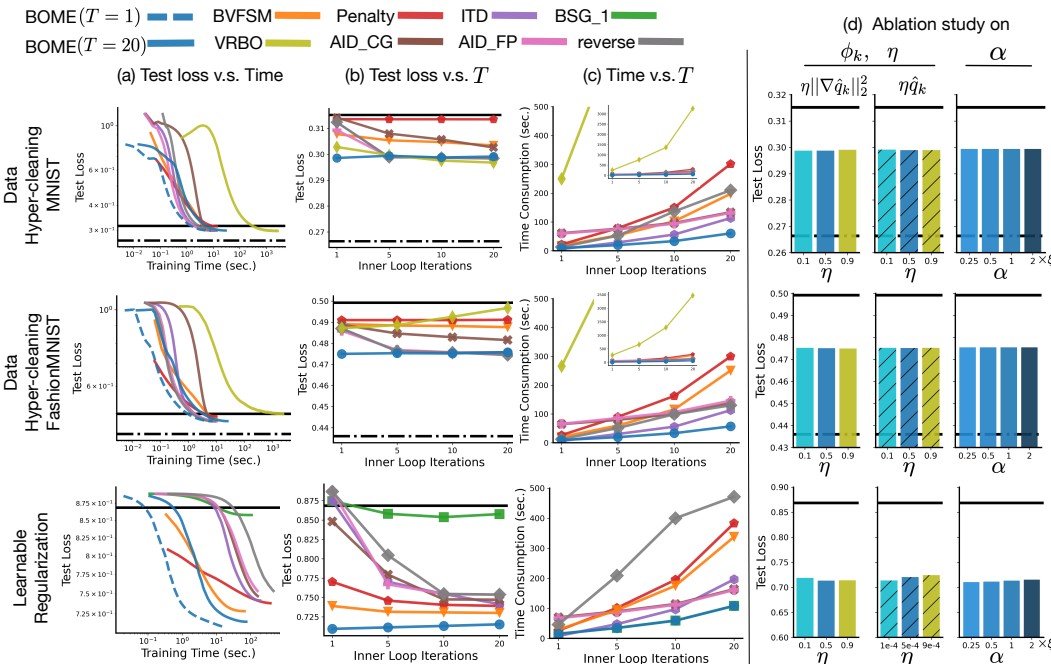

Figure 5: Bilevel optimization for hyperparameter optimization. **Top:** data hyper-cleaning on MNIST dataset. The solid black line is the model performance trained purely on the validation set and the dashed black line is the model performance trained on the validation set and on the part of training set that have correct labels. **Middle:** data hyper-cleaning on FashionMNIST dataset. **Bottom:** Learnable regularization on 20 Newsgroup dataset. The solid black line indicates the model performance without any regularization. The results for each method is averaged over 5 independent runs.

**Remark:** In Fig. 2 (top row), we do not include the performance of BSG-1 as we fail to find a set of hyperparameters for BSG-1 to make it work on these data hyper-cleaning problems. VRBO's performance at convergence is tuned by hyparameter search. However, we observe that VRBO learns slowly in practice, as it requires multiple steps of Hessian vector products at each step. We notice that this is slightly inconsistent with the findings in the original paper [51]. We adapt the code from https://github.com/JunjieYang97/MRVRBO and find the original implementation is also slow. It is possible that a good set of hyperparameters can result in better performance.

## A.5 Learnable regularization

The bilevel optimization formulation of the learnable rgularization problem is:

$$\min_{v,\theta} \ell^{\mathrm{val}}(\theta) \quad s.t. \quad \theta \in \arg\min_{\theta'} \ell^{\mathrm{train}}(\theta') + \|W_v \theta'\|_2^2.$$

We use a linear model who's parameter $\theta$ is a matrix (e.g. $\theta \in \mathbb{R}^{20 \times 130107}$). Hence $v \in \mathbb{R}^{130107}$. For this experiment, the inner stepsize $\alpha$ of all methods are searched from $\{1, 10, 100, 1000\}$. The outer stepsize $\xi$ is searched from $\{0.5, 1, 5, 10, 50, 100, 500, 1000\}$. For BVFSM and Penalty methods, we also search for their best hyperparameters from $\{0.001, 0.01, 0.1, 1\}$. Similar to the experiment on Data Hyper-cleaning, we split the dataset into 4 parts: train set, validation set 1, validation set 2, and the test set. The initial model parameter $\theta_0$ is initialized from a pretrained model without any regularization (e.g. $v = 0$) to speed up the learning. To conduct the ablation on $\alpha$ for BOME, we search for $\alpha \in \{0.25\xi, 0.5\xi, \xi, 2\xi\}$, where $\xi = 100$ is the best stepsize we found for BOME. In the bottom left of Fig. 5, we compare BOME with ($T = 1$ and $T = 20$) with baseline methods whose $T$ is chosen based on best performance on the validation set 2.

**Remark:** In Fig. 2 (bottom row), we do not include the performance of VRBO as we fail to find a set of hyperparameters for VRBO that works well on the learnable regularization experiment.

## A.6 Continual Learning

Continual learning (CL) experiment follows closely to the setup in contextual transformation network (CTN) from Pham et al. [43], which trains a deep neural network consisting of a quickly updated backbone network (parameterized by $\theta$) a slowly updated controller network (parameterized by $\theta$). When training the $\tau$-th task, the update on $(v, \theta)$ is solved from a bilevel optimization:

$$\min_{v,\theta} \ell_{1:\tau}^{\mathrm{val}}(v, \theta) \quad \text{s.t.} \quad \theta \in \arg\min_{\theta'} \ell_{1:\tau}^{\mathrm{train}}(v, \theta').$$

More specifically,

$$\ell_{1:\tau}^{\mathrm{val}}(v, \theta) = L^{\mathrm{ctrl}}(\{\theta^*(v), v\}; \mathcal{M}_{<t+1}^{\mathrm{sm}}), \quad \text{and} \quad \ell_{1:\tau}^{\mathrm{train}}(v, \theta') = L^{\mathrm{tr}}(\{\theta, v, \}, D_t \cup \mathcal{M}_{<t}^{\mathrm{em}}). \quad (14)$$

Here, $\mathcal{M}_t^{\mathrm{sm}}$ and $\mathcal{M}_t^{\mathrm{em}}$ denote the semantic and episodic memory of task $t$ (e.g. they can be think of validation and training data) and $D_t$ is the training data of task $t$. Hence, the inner objective learns a backbone $\theta^*(v)$ that performs well on the training data which consists of the current task data $D_t$ as well as previous episodic memories $\mathcal{M}_{<t}^{\mathrm{em}}$. Then, the outer objective encourages good generalization on the held out validation data, which consists of the semantic memory $\mathcal{M}_{<t+1}^{\mathrm{sm}}$. All hyperparameters of BOME are set to the same as those of CTN. We choose $\phi_k = \eta \|\nabla \hat{q}_k(v_k, \theta_k)\|$ where $\eta = 2.0$, here $\eta$ is chosen from $\{0.1, 0.5, 1.0, 2.0\}$.

| Method | PMNIST | | | Split CIFAR | | |
|---|---|---|---|---|---|---|
| | ACC (↑) | NBT (↓) | FT (↑) | ACC (↑) | NBT (↓) | FT (↑) |
| Offline | $84.95 \pm 0.95$ | - | - | $74.11 \pm 0.66$ | - | - |
| EWC | $48.07 \pm 1.67$ | $30.55 \pm 1.66$ | $78.62 \pm 0.60$ | $36.97 \pm 1.87$ | $29.61 \pm 2.32$ | $66.58 \pm 0.80$ |
| MER | $76.59 \pm 0.74$ | $6.88 \pm 0.59$ | $82.32 \pm 0.34$ | $60.32 \pm 0.86$ | $11.80 \pm 0.86$ | $69.23 \pm 0.40$ |
| GEM | $72.74 \pm 0.91$ | $7.79 \pm 1.04$ | $80.53 \pm 0.28$ | $61.33 \pm 1.16$ | $8.04 \pm 1.10$ | $69.37 \pm 0.72$ |
| ER-Ring | $72.11 \pm 0.46$ | $7.96 \pm 0.46$ | $80.06 \pm 0.37$ | $61.96 \pm 1.22$ | $7.18 \pm 1.84$ | $69.14 \pm 0.87$ |
| CTN (+ITD) | $78.40 \pm 0.28$ | $5.62 \pm 0.39$ | $84.02 \pm 0.29$ | $67.7 \pm 60.96$ | $4.88 \pm 0.77$ | $72.58 \pm 0.62$ |
| CTN (+BVFSM) | $77.78 \pm 0.32$ | $7.25 \pm 0.28$ | $\mathbf{85.03} \pm 0.28$ | $67.04 \pm 0.76$ | $6.97 \pm 0.62$ | $\mathbf{74.01} \pm 0.57$ |
| CTN (+Penalty) | $61.57 \pm 0.29$ | $12.69 \pm 0.44$ | $74.27 \pm 0.29$ | $47.41 \pm 2.93$ | $8.36 \pm 2.63$ | $73.82 \pm 0.64$ |
| CTN (+BOME) | $\mathbf{80.70} \pm 0.26$ | $\mathbf{4.09} \pm 0.27$ | $84.79 \pm 0.25$ | $\mathbf{68.16} \pm 0.60$ | $\mathbf{4.72} \pm 0.75$ | $72.88 \pm 0.48$ |

Table 2: Results of continual learning as bilevel optimization. We compute the mean and standard error of each method's results over 5 independent runs. Best results are **bolded**.

# B    Proof of the Result in Section 4.1

We proof Proposition 1 using Proposition 6.3 (presented below using our notation) in Gong et al. [16] by checking all the assumptions required by Proposition 6.3 in Gong et al. [16] are satisfied. Specifically, it remains to show that for any $k$, $\lambda_k < \infty$, $\lim_{k\to\infty} \nabla q(v_k, \theta_k) = 0$ and $q$ is lower bounded (this is trivial as $q \geq 0$ by its definition), which we prove below.

Firstly, simple calculation shows that for any $k$,

$$\lambda_k = \left[ \frac{-\langle \nabla f(v_k, \theta_k), \nabla q(v_k, \theta_k) \rangle}{||\nabla q(v_k, \theta_k)||^2} \right]_+ \leq \frac{\sup_{v,\theta} ||\nabla f(v,\theta)||}{||\nabla q(v_k, \theta_k)||} < \infty.$$

Here the last inequality is by $||\nabla q(v_k, \theta_k)|| > 0$. Secondly, note that as we assume $\nabla q$ is continuous, this implies that

$$\lim_{k\to\infty} \nabla q(v_k, \theta_k) = \nabla q(v^*, \theta^*).$$

As $q(v^*, \theta^*) = 0$, we have $\nabla q(v^*, \theta^*) = 0$. Using Proposition 6.3 in Gong et al. [16] gives the desired result.

**Proposition B.1** (Proposition 6.3 in Gong et al. [16]). *Assume $f, q, \nabla q$ are continuously differentiable. Let $\{[v_k, \theta_k, \lambda_k] : k = 1, 2, ...\}$ be a sequence which satisfies $\lim_{k\to\infty} ||\nabla q(v_k, \theta_k)|| = 0$ and $\lim_{k\to\infty} ||\nabla f(v_k, \theta_k) + \lambda_k \nabla q(v_k, \theta_k)|| = 0$. Assume that $[v^*, \theta^*]$ is a limit point of $[v_k, \theta_k]$ as $k \to \infty$ and $[v^*, \theta^*]$ satisfies CRCQ with $\nabla_\theta q$, then there exists a vector-valued Lagrange multiplier $\omega^* \in \mathbb{R}^m$ (the same length as $\theta$) such that*

$$\nabla f(v^*, \theta^*) + \nabla(\nabla_\theta q(v^*, \theta^*))\omega^* = 0.$$

# C    Proof of the Result in Section 4.2

We define $L_q := 2L(L/\kappa + 1)$ and using Assumption 2 and 1, we are able to show that $q(v, \theta)$ is $L_q$-smooth (see Lemma 4 for details). For simplicity, we also assume that $\xi \leq 1$ throughout the proof. We use $b$ with some subscript to denote some general $O(1)$ constant and refer reader to section E for their detailed value.

Note that $\hat{q}$ defined in Section 3 changes in different iterations (as it depends on $\theta_k^{(T)}$) and so does $\nabla \hat{q}$. To avoid the confusion, we introduce several new notations. Firstly, given $v$ and $\theta$, $\theta^{(T)}$ denotes the results of $T$ steps of gradient of $g(v, \cdot)$ w.r.t. $\theta$ starting from $\theta$ with step size $\alpha$ (similar to the definition in (8)). Note that $\theta^{(T)}$ depends on $v, \theta$ and $\alpha$. Our notation does not reflects this dependency on $v, \alpha$ as we find it introduces no ambiguity while much simplifies the notation. Also note that when taking gradient on $\hat{q}$, the $\theta_k^{(T)}$ at iteration $k$ is treated as a constant and the gradient does not pass through it. To be clear, we define $\hat{\nabla} q(v, \theta) = \nabla g(v, \theta) - \left[ \nabla_1^\top g(v, \theta^{(T)}), \mathbf{0}^\top \right]^\top$, where $\mathbf{0}$ denotes a zero vector with the same dimension as $\theta$. Using this definition, $\hat{\nabla} q(v_k, \theta_k) = \nabla \hat{q}(v_k, \theta_k)$ at iteration $k$. We also let $\lambda^*(v, \theta)$ be the solution of the dual problem of

$$\min_\delta ||\hat{\nabla} q(v, \theta) - \nabla f(v, \theta)||^2 \ s.t. \ \left\langle \hat{\nabla} q(v, \theta), \nabla f(v, \theta) \right\rangle \geq \eta ||\hat{\nabla} q(v, \theta)||^2. \tag{15}$$

That is

$$\lambda^*(v, \theta) = \begin{cases} \frac{[\eta ||\hat{\nabla} q(v,\theta)||^2 - \langle \hat{\nabla} q(v,\theta), \nabla f(v,\theta) \rangle]_+}{||\hat{\nabla} q(v,\theta)||^2} & \text{when } ||\hat{\nabla} q(v, \theta)|| > 0 \\ 0 & \text{when } ||\hat{\nabla} q(v, \theta)|| = 0 \end{cases} \tag{16}$$

We might use $\lambda^*$ for $\lambda^*(v, \theta)$ when it introduces no confusion. Also, denote $\delta^*(v, \theta) = \lambda^*(v, \theta)\hat{\nabla} q(v, \theta) + \nabla f(v, \theta)$ and thus $\delta_k = \delta^*(v_k, \theta_k)$.

We start with several technical Lemmas showing some basic function properties.

## C.1    Technical Lemmas

**Lemma 1.** *Under Assumption 1, for any $v, \theta$, $g(v, \theta) - g(v, \theta^*(v)) \geq \frac{\kappa}{4} ||\theta - \theta^*(v)||^2$.*

**Lemma 2.** *Under Assumption 1 and 2, we have* $||\nabla q(v,\theta) - \hat{\nabla} q(v,\theta)|| \leq L||\theta^{(T)} - \theta^*(v)||$ *for any* $v, \theta$. *Also, when* $||\hat{\nabla} q(v,\theta)|| = 0$, $q(v,\theta) = 0$.

**Lemma 3.** *Under Assumption 2, 1, we have* $||\theta^*(v_2) - \theta^*(v_1)|| \leq \frac{2L}{\kappa}||v_1 - v_2||$.

**Lemma 4.** *Under Assumption 2, we have* $||\nabla_\theta q(v,\theta_1) - \nabla_\theta q(v,\theta_2)|| \leq L||\theta_1 - \theta_2||$, *for any* $v$. *Further assume Assumption 1, we have*

$$||\nabla q(v_1,\theta_1) - \nabla q(v_2,\theta_2)|| \leq L_q||[v_1,\theta_1] - [v_2,\theta_2]||,$$

*where* $L_q := 2L(L/\kappa + 1)$.

**Lemma 5.** *Under Assumption 1, 2 and assume that* $\alpha < 2/L$. *Given any* $v, \theta$, *suppose* $\theta^{(0)} = \theta$ *and* $\theta^{(t+1)} = \theta^{(t)} - \alpha\nabla_\theta q(v,\theta^{(t)})$, *then for any* $t$, *we have* $q(v,\theta^{(t)}) \leq \exp(-b_1(\alpha, L, \kappa)t)q(v,\theta)$, *where* $b_1(\alpha, L, \kappa) =$ *is some strictly positive constant that depends on* $\alpha$, $L$ *and* $\kappa$.

**Lemma 6.** *Under Assumption 3, for any* $[v,\theta]$, *we have* $||\delta^*(v,\theta)||, ||\nabla q(v,\theta)||, ||\hat{\nabla} q(v,\theta)|| \leq b_2(M,\eta)$, *where* $b_2(M,\eta) = (3+\eta)M$.

**Lemma 7.** *Under Assumption 3, for any* $[v,\theta]$, *we have* $\lambda^*||\hat{\nabla} q||^2 \leq \eta||\hat{\nabla} q||^2 + M||\hat{\nabla} q||$, *where* $\lambda^*$ *are defined in (16)*.

**Lemma 8.** *Under Assumption 1 and 2, we have* $||\nabla q(v,\theta)|| \leq 2\kappa^{-1/2}L_q q^{1/2}(v,\theta)$.

### C.1.1 Lemmas

Now we give several main lemmas that are used to prove the result in Section 4.2.

**Lemma 9.** *Under Assumption 1, 2 and 3, when* $||\hat{\nabla} q(v_k,\theta_k)|| > 0$, *we have*

$$q(v_{k+1},\theta_{k+1}) - q(v_k,\theta_k) \leq -\xi\eta||\nabla q(v_k,\theta_k)||^2 + \xi\eta L_q||\theta_k^{(T)} - \theta^*(v_k)|| \left(L_q||\theta_k^{(T)} - \theta^*(v_k)|| + 2L_q||\theta_k - \theta^*(v_k)||\right)$$
$$+ \xi b_2 L||\theta_k^{(T)} - \theta^*(v_k)|| + L_q\xi^2 b_2^2/2.$$

*When* $||\hat{\nabla} q(v_k,\theta_k)|| = 0$, *we have* $q(v_{k+1},\theta_{k+1}) - q(v_k,\theta_k) \leq \xi^2 L_q b_2^2/2$.

**Lemma 10.** *Under Assumption 1, 2 and 3, choosing* $T \geq b_3(\eta, r, \kappa, L)$, *we have*

$$q(v_k,\theta_k) \leq \exp(-b_4 k)q(v_0,\theta_0) + \Delta,$$

*where* $b_4 = -\log(1 - \frac{\xi}{4}\eta\kappa)$ *is some strictly positive constant and* $\Delta = O(\exp(-b_1 T) + \xi)$.

**Lemma 11.** *Under Assumption 1, 2 and 3, we have*

$$\sum_{k=0}^{K-1} ||\nabla q(v_k,\theta_k)||^2 \leq \frac{b_5 q(v_0,\theta_0)}{\xi} + K\xi^2 b_6 \Delta,$$

*where* $b_5$ *is some constant depends on* $L_q, \eta, \kappa$; $b_6$ *is some constant depends on* $\kappa, L$ *and* $\Delta$ *is defined in Lemma 10*.

**Lemma 12.** *Under Assumption 1, 2 and 3, choosing* $T \geq b_3(\eta, r, \kappa, L)$ *and assume that* $r, \xi \leq 1/L$, *we have*

$$\sum_{k=0}^{K-1} \left[||\delta^*(v_k,\theta_k)||^2 + q(v_k,\theta_k)\right] = O(\xi^{-1} + K\exp(-b_1 T/2) + K\xi^{1/2} + \xi^{-1/2}K^{1/2}q^{1/2}(v_0,\theta_0)).$$

### C.2 Proof of Theorem 1

Using our definition of $\lambda^*$ in (16), we have

$$||\nabla f(v,\theta) + \lambda^*(v,\theta)\nabla q(v,\theta)|| \leq ||\nabla f(v,\theta) + \lambda^*(v,\theta)\hat{\nabla} q(v,\theta)|| + ||\lambda^*(v,\theta)(\hat{\nabla} q(v,\theta) - \nabla q(v,\theta))||$$
$$= ||\delta^*(v,\theta)|| + ||\lambda^*(v,\theta)(\hat{\nabla} q(v,\theta) - \nabla q(v,\theta))||.$$

Using Lemma 2, we know that when $||\hat{\nabla} q|| = 0$, we have $q = 0$ and thus $||\nabla q|| = 0$. In this case, $||\lambda^*(\hat{\nabla} q - \nabla q)|| = 0$. When $||\hat{\nabla} q|| > 0$, some algebra shows that

$$||\lambda^*(\hat{\nabla} q - \nabla q)|| \leq \left[\eta - \left\langle \nabla f, \hat{\nabla} q / ||\hat{\nabla} q|| \right\rangle ||\hat{\nabla} q||^{-1}\right] ||\hat{\nabla} q - \nabla q||$$
$$\leq (\eta - \left\langle \nabla f, \hat{\nabla} q / ||\hat{\nabla} q|| \right\rangle ||\hat{\nabla} q||^{-1}) ||\hat{\nabla} q - \nabla q||.$$

Notice that

$$||\hat{\nabla} q(v, \theta) - \nabla q(v, \theta)|| \leq L||\theta^{(T)} - \theta^*(v)||$$
$$\leq 2L\kappa^{-1/2} q^{1/2}(v, \theta^{(T)})$$
$$\leq 2L\kappa^{-1/2} \exp(-b_1 T/2) q^{1/2}(v, \theta)$$
$$\leq 2L\kappa^{-1} \exp(-b_1 T/2) ||\nabla q(v, \theta)||.$$

Here the first inequality is by Lemma 2, the second inequality is by Lemma 1, the third inequality is by Lemma 5 and the last inequality is by Assumption 1 (using $||\nabla q(v, \theta)|| \geq ||\nabla_\theta g(v, \theta)||$). Similarly, under assumption that $T \geq \lceil -b_1^{-1} \log(\frac{1}{16} \kappa^2 L^{-2}) \rceil$, $L\kappa^{-1} \exp(-b_1 T/2) \leq 1/4$,

$$||\hat{\nabla} q(v, \theta)|| = ||\hat{\nabla} q(v, \theta) - \nabla q(v, \theta) + \nabla q(v, \theta)||$$
$$\geq ||\nabla q(v, \theta)|| - ||\hat{\nabla} q(v, \theta) - \nabla q(v, \theta)||$$
$$\geq ||\nabla q(v, \theta)||(1 - (2L\kappa^{-1} \exp(-b_1 T/2)))$$
$$\geq \frac{1}{2} ||\nabla q(v, \theta)||.$$

This implies that

$$\frac{||\hat{\nabla} q - \nabla q||}{||\hat{\nabla} q||} \leq 2 \frac{||\hat{\nabla} q - \nabla q||}{||\nabla q||} \leq 4L\kappa^{-1} \exp(-b_1 T/2).$$

We thus have

$$||\lambda^*(v, \theta)(\hat{\nabla} q(v, \theta) - \nabla q(v, \theta))|| \leq \eta ||\hat{\nabla} q - \nabla q|| + \left\langle \nabla f, \frac{\hat{\nabla} q}{||\hat{\nabla} q||} \right\rangle \frac{||\hat{\nabla} q - \nabla q||}{||\hat{\nabla} q||}$$
$$\leq 2L\kappa^{-1} \exp(-b_1 T/2) \left[\eta ||\nabla q(v, \theta)|| + 2 \left\langle \nabla f, \frac{\hat{\nabla} q}{||\hat{\nabla} q||} \right\rangle \right]$$
$$\leq 2L\kappa^{-1} \exp(-b_1 T/2)(\eta + 2) b_2,$$

where the last inequality is by Lemma 6. Combining all the results and using $||\nabla q(v_k, \theta_k)|| \leq 2\kappa^{-1/2} L_q q^{1/2}(v_k, \theta_k)$ by Lemma 8, we have

$$\mathcal{K}(v, \theta) \leq ||\nabla f(v, \theta) + \lambda^*(v, \theta) \nabla q(v, \theta)||^2 + q(v, \theta)$$
$$\leq 2||\nabla f(v, \theta) + \lambda^*(v, \theta) \hat{\nabla} q(v, \theta)||^2 + q(v, \theta) + 2||\lambda^*(v, \theta)(\hat{\nabla} q(v, \theta) - \nabla q(v, \theta))||^2$$
$$\leq 2||\delta^*(v, \theta)||^2 + q(v, \theta) + 8L^2 \kappa^{-2} \exp(-b_1 T)(\eta + 2)^2 b_2^2.$$

Using Lemma 12, we have

$$\min_k \mathcal{K}(v_k, \theta_k) = O(\min_k(||\delta^*(v_k, \theta_k)||^2 + q(v_k, \theta_k)) + \exp(-b_1 T))$$
$$= O(\xi^{-1} + K \exp(-b_1 T/2) + K\xi^{1/2} + \xi^{-1/2} K^{1/2} q^{1/2}(v_0, \theta_0)).$$

### C.3 Proof of Lemmas

#### C.3.1 Proof of Lemma 9

When $||\hat{\nabla}q(v_k, \theta_k)|| > 0$, by Lemma 4, we know that $q$ is $L_q$-smoothness, we have

$$q(v_{k+1}, \theta_{k+1}) - q(v_k, \theta_k) \leq -\xi \langle \nabla q(v_k, \theta_k), \delta^*(v_k, \theta_k) \rangle + \frac{L_q \xi^2}{2} ||\delta^*(v_k, \theta_k)||^2$$

$$\leq -\xi \langle \hat{\nabla}q(v_k, \theta_k), \delta^*(v_k, \theta_k) \rangle - \xi \langle \nabla q(v_k, \theta_k) - \hat{\nabla}q(v_k, \theta_k), \delta^*(v_k, \theta_k) \rangle + L_q \xi^2 b_2^2 / 2$$

$$\leq -\xi\eta ||\hat{\nabla}q(v_k, \theta_k)||^2 - \xi \langle \nabla q(v_k, \theta_k) - \hat{\nabla}q(v_k, \theta_k), \delta^*(v_k, \theta_k) \rangle + L_q \xi^2 b_2^2 / 2$$

$$\leq -\xi\eta ||\hat{\nabla}q(v_k, \theta_k)||^2 + \xi b_2 ||\nabla q(v_k, \theta_k) - \hat{\nabla}q(v_k, \theta_k)|| + L_q \xi^2 b_2^2 / 2.$$

where the second and the last inequality is by Lemma 6 and the third inequality is ensured by the constraint in the local subproblem ($\langle \nabla \hat{\nabla}q(v_k, \theta_k), \delta^*(v_k, \theta_k) \rangle \geq \eta ||\hat{\nabla}q(v_k, \theta_k)^2||.$). And by Lemma 2, we have $||\nabla q(v_k, \theta_k) - \hat{\nabla}q(v_k, \theta_k)|| \leq L||\theta_k^{(T)} - \theta^*(v_k)||$. Plug in the bound we have

$$q(v_{k+1}, \theta_{k+1}) - q(v_k, \theta_k) \leq -\xi\eta ||\hat{\nabla}q(v_k, \theta_k)||^2 + \xi b_2 L ||\theta_k^{(T)} - \theta^*(v_k)|| .$$

Also notice that

$$\left| ||\hat{\nabla}q(v_k, \theta_k)||^2 - ||\nabla q(v_k, \theta_k)||^2 \right| \leq ||\hat{\nabla}q(v_k, \theta_k) - \nabla q(v_k, \theta_k)|| \, ||\hat{\nabla}q(v_k, \theta_k) + \nabla q(v_k, \theta_k)||$$

$$\leq ||\hat{\nabla}q(v_k, \theta_k) - \nabla q(v_k, \theta_k)|| \, (||\hat{\nabla}q(v_k, \theta_k) - \nabla q(v_k, \theta_k)|| + 2||\nabla q(v_k, \theta_k)||)$$

$$\leq L_q ||\theta_k^{(T)} - \theta^*(v_k)|| \, (L_q ||\theta_k^{(T)} - \theta^*(v_k)|| + 2||\nabla q(v_k, \theta_k)||)$$

$$= L_q ||\theta_k^{(T)} - \theta^*(v_k)|| \, (L_q ||\theta_k^{(T)} - \theta^*(v_k)|| + 2||\nabla q(v_k, \theta_k) - \nabla q(v_k, \theta^*(v_k))||)$$

$$\leq L_q ||\theta_k^{(T)} - \theta^*(v_k)|| \, (L_q ||\theta_k^{(T)} - \theta^*(v_k)|| + 2L_q ||\theta_k - \theta^*(v_k)||),$$

where the third inequality is by Lemma 2, the equality is by $\nabla q(v_k, \theta^*(v_k)) = 0$ and the last inequality is by Lemma 4.

Using this bound, we further have

$$q(v_{k+1}, \theta_{k+1}) - q(v_k, \theta_k) \leq -\xi\eta ||\nabla q(v_k, \theta_k)||^2 + \xi\eta \left| ||\hat{\nabla}q(v_k, \theta_k)||^2 - ||\nabla q(v_k, \theta_k)||^2 \right|$$

$$+ \xi b_2 ||\nabla q(v_k, \theta_k) - \hat{\nabla}q(v_k, \theta_k)|| + L_q \xi^2 b_2^2 / 2$$

$$\leq -\xi\eta ||\nabla q(v_k, \theta_k)||^2 + \xi\eta L_q ||\theta_k^{(T)} - \theta^*(v_k)|| \, (L_q ||\theta_k^{(T)} - \theta^*(v_k)|| + 2L_q ||\theta_k - \theta^*(v_k)||)$$

$$+ \xi b_2 L ||\theta_k^{(T)} - \theta^*(v_k)|| + L_q \xi^2 b_2^2 / 2.$$

When $||\hat{\nabla}q(v_k, \theta_k)|| = 0$, by Lemma 2, $q(v_k, \theta_k) = 0$ and hence $\nabla q(v_k, \theta_k) = 0$. We thus have

$$q(v_{k+1}, \theta_{k+1}) - q(v_k, \theta_k) \leq -\xi \langle \nabla q(v_k, \theta_k), \delta^*(v_k, \theta_k) \rangle + \frac{L_q \xi^2}{2} ||\delta^*(v_k, \theta_k)||^2$$

$$= \frac{L_q \xi^2}{2} ||\delta^*(v_k, \theta_k)||^2$$

$$\leq \xi^2 L_q b_2^2 / 2.$$

#### C.3.2 Proof of Lemma 10

By Lemma 9, when $||\hat{\nabla}q(v_k, \theta_k)|| > 0$, we have

$$q(v_{k+1}, \theta_{k+1}) - q(v_k, \theta_k) \leq -\xi\eta ||\nabla q(v_k, \theta_k)||^2 + \xi\eta L_q ||\theta_k^{(T)} - \theta^*(v_k)|| \, (L_q ||\theta_k^{(T)} - \theta^*(v_k)|| + 2L_q ||\theta_k - \theta^*(v_k)||)$$

$$+ \xi b_2 L ||\theta_k^{(T)} - \theta^*(v_k)|| + L_q \xi^2 b_2^2 / 2.$$

By Lemma 1 and Lemma 5

$$||\theta_k^{(T)} - \theta^*(v_k)|| \leq 2\kappa^{-1/2} q^{1/2}(v_k, \theta_k^{(T)}) \leq 2\kappa^{-1/2} \exp(-b_1 T/2) q^{1/2}(v_k, \theta_k).$$

$$||\theta_k - \theta^*(v_k)|| \leq 2\kappa^{-1/2} q^{1/2}(v_k, \theta_k).$$

Using those bounds, we know that

$$L_q||\theta_k^{(T)} - \theta^*(v_k)|| \; (L_q||\theta_k^{(T)} - \theta^*(v_k)|| + 2L_q||\theta_k - \theta^*(v_k)||) \leq 12L_q^2\kappa^{-1}\exp(-b_1T)q(v_k, \theta_k)$$

This implies that

$$\begin{aligned}
&q(v_{k+1}, \theta_{k+1}) - q(v_k, \theta_k)\\
&\leq - \xi\eta||\nabla q(v_k, \theta_k)||^2 + 12\xi\eta L_q^2\kappa^{-1}\exp(-b_1T)q(v_k, \theta_k)\\
&+ 2\xi b_2 L\kappa^{-1/2}\exp(-b_1T/2)q^{1/2}(v_k, \theta_k) + L_q\xi^2 b_2^2/2\\
&\leq - \xi\eta\kappa q(v_k, \theta_k) + 12\xi\eta L_q^2\kappa^{-1}\exp(-b_1T)q(v_k, \theta_k)\\
&+ 2\xi b_2 L\kappa^{-1/2}\exp(-b_1T/2)q^{1/2}(v_k, \theta_k) + L_q\xi^2 b_2^2/2.
\end{aligned}$$

Choosing $T$ such that $T \geq b_3(\eta, \alpha, \kappa, L)$ where

$$b_3(\eta, \alpha, \kappa, L) = \left\lceil -b_1^{-1}\log(\frac{\eta\kappa}{64\eta L_q^2}) \right\rceil,$$

we have

$$q(v_{k+1}, \theta_{k+1}) - q(v_k, \theta_k) \leq -\frac{3}{4}\xi\eta\kappa q(v_k, \theta_k) + 2\xi b_2 L\kappa^{-1/2}\exp(-b_1T/2)q^{1/2}(v_k, \theta_k) + L_q\xi^2 b_2^2/2.$$

This implies that when $\frac{64b_2^2 L^2}{\eta^2\kappa}\exp(-b_1T) \leq q(v_k, \theta_k)$ and $\frac{2L_q\xi b_2^2}{\eta\kappa} \leq q(v_k, \theta_k)$,

$$q(v_{k+1}, \theta_{k+1}) - q(v_k, \theta_k) \leq -\frac{1}{4}\xi\eta\kappa q(v_k, \theta_k).$$

Let $a = \max(\frac{64b_2^2 L^2}{\eta^2\kappa}\exp(-b_1T), \frac{2L_q\xi b_2^2}{\eta\kappa})$. Also, when $q(v_k, \theta_k) < a$,

$$\begin{aligned}
q(v_{k+1}, \theta_{k+1}) &\leq q(v_k, \theta_k) + 2\xi b_2 L\kappa^{-1/2}\exp(-b_1T/2)q^{1/2}(v_k, \theta_k) + L_q\xi^2 b_2^2/2\\
&< a + 2\xi b_2 L\kappa^{-1/2}\exp(-b_1T/2)\sqrt{a} + L_q\xi^2 b_2^2/2.
\end{aligned}$$

Note that

$$2\xi b_2 L\kappa^{-1/2}\exp(-b_1T/2) \leq \frac{\xi\eta\kappa}{4}\sqrt{a}$$

$$L_q\xi^2 b_2^2/2 \leq \frac{\xi\eta\kappa}{4}a.$$

This gives that in the case of $q(v_k, \theta_k) < a$,

$$q(v_{k+1}, \theta_{k+1}) < (1 + \frac{\xi\eta\kappa}{4})a.$$

Define $k_0$ as the first iteration such that $q(v_k, \theta_k) < a$. This implies that, for any $k \leq k_0$,

$$q(v_k, \theta_k) \leq (1 - \frac{\xi}{4}\eta\kappa)^k q(v_0, \theta_0).$$

When any $k > k_0$, we show that $q(v_{k+1}, \theta_{k+1}) \leq (1 + \frac{\xi\eta\kappa}{4})a$. This can be proved by induction. At $k = k_0 + 1$, if $q(v_k, \theta_k) < a$, we have $q(v_k, \theta_k) < (1 + \frac{\xi\eta\kappa}{4})a$. Else if at $k = k_0 + 1$, $q(v_k, \theta_k) \geq a$, $q(v_{k+1}, \theta_{k+1}) \leq q(v_k, \theta_k) \leq a$. We thus have the conclusion that for any $k > k_0$, $q(v_k, \theta_k) \leq (1 + \frac{\eta\kappa}{4})a$. Combining the result, we have

$$q(v_k, \theta_k) \leq (1 - \frac{\xi}{4}\eta\kappa)^k q(v_0, \theta_0) + \Delta,$$

where we denote

$$\Delta = (1 + \frac{\eta\kappa}{4})(\frac{64b_2^2 L^2}{\eta^2\kappa^3}\exp(-b_1T) + \frac{2L_q\xi b_2^2}{\eta\kappa}) + L_q\xi^2 b_2^2/2 = O(\exp(-b_1T) + \xi). \quad (17)$$

Let $b_4(\eta, \kappa, \xi) = -\log(1 - \frac{\xi}{4}\eta\kappa)$, we have the desired result.

### C.3.3   Proof of Lemma 11

By Lemma 8 and 10, we have

$$||\nabla q(v_k, \theta_k)||^2 \le 2\kappa^{-1}L_q^2 q(v_k, \theta_k)$$
$$\le 2\kappa^{-1}L_q^2\left[\exp(-b_4 k)q(v_0, \theta_0) + \Delta\right],$$

where $\Delta$ is defined in (17). Also notice that

$$||\hat{\nabla} q(v, \theta)|| \le ||\hat{\nabla} q(v, \theta) - \nabla q(v, \theta)|| + ||\nabla q(v, \theta)||$$
$$\le L||\theta^{(T)} - \theta^*(v)|| + ||\nabla q(v, \theta)||$$
$$\le 2L\kappa^{-1/2}q^{1/2}(v, \theta^{(T)}) + ||\nabla q(v, \theta)||$$
$$\le 2L\kappa^{-1/2}\exp(-b_1 T/2)q^{1/2}(v, \theta) + ||\nabla q(v, \theta)||$$
$$\le (2L\kappa^{-1}\exp(-b_1 T/2) + 1)||\nabla q(v, \theta)||$$
$$\le (2L\kappa^{-1} + 1)||\nabla q(v, \theta)||$$

Here the first inequality is by triangle inequality, the second inequality is by Lemma 2, the third inequality is by Lemma 1, the forth inequality is by Lemma 5 and the fifth inequality is by Assumption 1. Taking summation over iteration and using Lemma 10, we have

$$\sum_{k=0}^{K-1}||\hat{\nabla} q(v, \theta)||^2 \le (2L\kappa^{-1} + 1)^2\sum_{k=0}^{K-1}||\nabla q(v_k, \theta_k)||^2$$
$$\le (2L\kappa^{-1} + 1)^2\left[2\kappa^{-1}L_q^2 q(v_0, \theta_0)\sum_{k=0}^{K-1}[\exp(-b_4 k)] + K\Delta\right]$$
$$\le (2L\kappa^{-1} + 1)^2\left[\frac{2\kappa^{-1}L_q^2 q(v_0, \theta_0)}{1 - \exp(-b_4)} + K\Delta\right]$$
$$= \frac{b_5 q(v_0, \theta_0)}{\xi} + Kb_6\Delta,$$

where we define $b_5(L_q, \eta, \kappa) = \frac{16L_q^2}{\eta\kappa^2}(2L\kappa^{-1} + 1)^2$ and $b_6(\kappa, L) = (2L\kappa^{-1} + 1)^2$.

### C.4   Proof of Lemma 12

Remind that by our definition of $\lambda^*$ in (16) and Assumption 2, we have

$$f(v_{k+1}, \theta_{k+1}) - f(v_k, \theta_k) \le -\xi\langle\nabla f(v_k, \theta_k), \delta^*(v_k, \theta_k)\rangle + \frac{L\xi^2}{2}||\delta^*(v_k, \theta_k)||^2$$
$$= -\xi\left\langle\delta^*(v_k, \theta_k) - \lambda^*(v_k, \theta_k)\hat{\nabla} q(v_k, \theta_k), \delta^*(v_k, \theta_k)\right\rangle + \frac{L\xi^2}{2}||\delta^*(v_k, \theta_k)||^2$$
$$= -(\xi - \frac{L\xi^2}{2})||\delta^*(v_k, \theta_k)||^2 + \xi\lambda^*(v_k, \theta_k)\left\langle\hat{\nabla} q(v_k, \theta_k), \delta^*(v_k, \theta_k)\right\rangle$$
$$\le -(\xi - \frac{L\xi^2}{2})||\delta^*(v_k, \theta_k)||^2 + \xi\eta\lambda^*(v_k, \theta_k)||\hat{\nabla} q(v_k, \theta_k)||^2$$
$$\le -\frac{\xi}{2}||\delta^*(v_k, \theta_k)||^2 + \xi\eta\lambda^*(v_k, \theta_k)||\hat{\nabla} q(v_k, \theta_k)||^2,$$

where the last inequality is by the assumption on $\xi \le 1/L$. To show the second inequality, we use the complementary slackness of Problem (15), that is

$$\lambda^*(v_k, \theta_k)\left[\left\langle\hat{\nabla} q(v_k, \theta_k), \delta^*(v_k, \theta_k)\right\rangle - \eta||\hat{\nabla} q(v_k, \theta_k)||\right] = 0.$$

By telescoping,

$$\sum_{k=0}^{K-1} f(v_{k+1}, \theta_{k+1}) - f(v_k, \theta_k) \leq -\frac{\xi}{2} \sum_{k=0}^{K-1} ||\delta^*(v_k, \theta_k)||^2 + \xi\eta \sum_{k=0}^{K-1} \lambda^*(v_k, \theta_k)||\hat{\nabla}q(v_k, \theta_k)||^2$$

$$\leq -\frac{\xi}{2} \sum_{k=0}^{K-1} ||\delta^*(v_k, \theta_k)||^2 + \xi\eta \sum_{k=0}^{K-1} (\eta||\hat{\nabla}q(v_k, \theta_k)||^2 + M||\hat{\nabla}q(v_k, \theta_k)||)$$

$$= -\frac{\xi}{2} \sum_{k=0}^{K-1} ||\delta^*(v_k, \theta_k)||^2 + \xi\eta^2 \sum_{k=0}^{K-1} ||\hat{\nabla}q(v_k, \theta_k)||^2 + \xi\eta M \sum_{k=0}^{K-1} ||\hat{\nabla}q(v_k, \theta_k)||$$

$$\leq -\frac{\xi}{2} \sum_{k=0}^{K-1} ||\delta^*(v_k, \theta_k)||^2 + \xi\eta^2 \sum_{k=0}^{K-1} ||\hat{\nabla}q(v_k, \theta_k)||^2 + \xi\eta M \sqrt{K} \sqrt{\sum_{k=0}^{K-1} ||\hat{\nabla}q(v_k, \theta_k)||^2},$$

where the second inequality is by Lemma 7 and the last inequality is by Holder's inequality. Since $\sum_{k=0}^{K-1} f(v_{k+1}, \theta_{k+1}) - f(v_k, \theta_k) = f(v_K, \theta_K) - f(v_0, \theta_0)$, rearrange the terms, we have

$$\xi \sum_{k=0}^{K-1} ||\delta^*(v_k, \theta_k)||^2 \leq 2(f(v_0, \theta_0) - f(v_K, \theta_K)) + 2\xi\eta^2 \sum_{k=0}^{K-1} ||\hat{\nabla}q(v_k, \theta_k)||^2 + 2\xi\eta M \sqrt{K} \sqrt{\sum_{k=0}^{K-1} ||\hat{\nabla}q(v_k, \theta_k)||^2}.$$

This implies that

$$\xi \sum_{k=0}^{K-1} \left[ ||\delta^*(v_k, \theta_k)||^2 + q(v_k, \theta_k) \right] \leq 2(f(v_0, \theta_0) - f(v_K, \theta_K)) + 2\xi\eta^2 \sum_{k=0}^{K-1} ||\hat{\nabla}q(v_k, \theta_k)||^2$$

$$+ 2\xi\eta M \sqrt{K} \sqrt{\sum_{k=0}^{K-1} ||\hat{\nabla}q(v_k, \theta_k)||^2} + \xi \sum_{k=0}^{K-1} q(v_k, \theta_k).$$

Using Lemma 10, we know that

$$q(v_k, \theta_k) \leq (1 - \frac{\xi}{4}\eta\kappa)^k q(v_0, \theta_0) + \Delta.$$

This gives that

$$\xi \sum_{k=0}^{K-1} q(v_k, \theta_k) \leq \frac{4q(v_0, \theta_0)}{\eta\kappa} + \xi K \Delta.$$

Using Lemma 11, 10 and $\sqrt{x+y} \leq \sqrt{x} + \sqrt{y}$, we have

$$2\xi\eta^2 \sum_{k=1}^{K} ||\hat{\nabla}q(v_k, \theta_k)||^2 \leq 2\eta^2 b_5 q(v_0, \theta_0) + 2K\eta^2\xi b_6 \Delta$$

$$2\xi\eta M \sqrt{K} \sqrt{\sum_{k=1}^{K} ||\hat{\nabla}q(v_k, \theta_k)||^2} \leq 2\xi^{1/2} K^{1/2} b_5^{1/2} \eta M q^{1/2}(v_0, \theta_0) + 2K\xi b_6^{1/2} \eta M \Delta^{1/2}$$

This implies that

$$\xi \sum_{k=0}^{K-1} \left[ ||\delta^*(v_k, \theta_k)||^2 + q(v_k, \theta_k) \right]$$

$$\leq 2(f(v_0, \theta_0) - f(v_K, \theta_K)) + 2\eta^2 b_5 q(v_0, \theta_0) + 2K\eta^2\xi b_6 \Delta + 2\xi^{1/2} K^{1/2} b_5^{1/2} \eta M q^{1/2}(v_0, \theta_0)$$

$$+ 2K\xi b_6^{1/2} \eta M \Delta^{1/2} + \frac{4q(v_0, \theta_0)}{\eta\kappa} + \xi K \Delta$$

$$\leq 2(f(v_0, \theta_0) - f(v_K, \theta_K)) + (2\eta^2 b_5 + \frac{4}{\eta\kappa})q(v_0, \theta_0) + 2K\xi(b_6^{1/2}\eta M \Delta^{1/2} + (b_6\eta^2 + 1/2)\Delta)$$

$$+ 2\xi^{1/2} K^{1/2} b_5^{1/2} \eta M q^{1/2}(v_0, \theta_0)$$

We thus have

$$\sum_{k=0}^{K-1} \left[ ||\delta^*(v_k, \theta_k)||^2 + q(v_k, \theta_k) \right]$$

$$= O(\xi^{-1} + K\Delta^{1/2} + \xi^{-1/2}K^{1/2}q^{1/2}(v_0, \theta_0))$$

$$= O(\xi^{-1} + K\exp(-b_1 T/2) + K\xi^{1/2} + \xi^{-1/2}K^{1/2}q^{1/2}(v_0, \theta_0)).$$

## C.5 Proofs of Technical Lemmas

### C.5.1 Proof of Lemma 1

Please see the proof of Theorem 2 in Karimi et al. [24].

### C.5.2 Proof of Lemma 2

Since $\nabla_2 g(v, \theta^*(v)) = 0$, we have $\nabla_v g(v, \theta^*(v)) = \nabla_1 g(v, \theta^*(v)) + \nabla_v \theta^*(v)\nabla_2 g(v, \theta^*(v)) = \nabla_1 g(v, \theta^*(v))$. Thus

$$\nabla q(v, \theta) = \left[ \begin{array}{c} \nabla_v g(v, \theta) - \nabla_v g(v, \theta^*(v)) \\ \nabla_\theta g(v, \theta) \end{array} \right] = \left[ \begin{array}{c} \nabla_v g(v, \theta) - \nabla_1 g(v, \theta^*(v)) \\ \nabla_\theta g(v, \theta) \end{array} \right].$$

Also note that

$$\hat{\nabla} q(v, \theta) = \left[ \begin{array}{c} \nabla_v g(v, \theta) - \nabla_1 g(v, \theta^{(T)}) \\ \nabla_\theta g(v, \theta) \end{array} \right].$$

This gives that

$$||\nabla q(v, \theta) - \hat{\nabla} q(v, \theta)|| = ||\nabla_1 g(v, \theta^{(T)}) - \nabla_1 g(v, \theta^*(v))||$$
$$\leq L||\theta^{(T)} - \theta^*(v)||.$$

Also when $0 = ||\hat{\nabla} q(v, \theta)|| = \sqrt{||\nabla_v g(v, \theta) - \nabla_1 g(v, \theta^{(T)})||^2 + ||\nabla_\theta g(v, \theta)||^2}$, we have $||\nabla_\theta g(v, \theta)|| = 0$. Under Assumption 1,

$$0 = ||\nabla_\theta g(v, \theta)|| \geq \kappa(g(v, \theta) - g(v, \theta^*(v))) = \kappa q(v, \theta).$$

### C.5.3 Proof of Lemma 3

Using Assumption 1 and $\nabla_2 g(v_1, \theta^*(v_1)) = 0$, we have

$$||\nabla_2 g(v_1, \theta^*(v_2))|| \geq \sqrt{\kappa(g(v_1, \theta^*(v_2)) - g(v_1, \theta^*(v_1)))}.$$

Also by Lemma 1, we have $g(v_1, \theta^*(v_2)) - g(v_1, \theta^*(v_1) \geq \frac{1}{4}\kappa||\theta^*(v_2) - \theta^*(v_1)||^2$. These imply that

$$||\nabla_2 g(v_1, \theta^*(v_2))|| \geq \frac{1}{2}\kappa||\theta^*(v_2) - \theta^*(v_1)||.$$

Also

$$||\nabla_2 g(v_1, \theta^*(v_2))||$$
$$= ||\nabla_2 g(v_1, \theta^*(v_2)) - \nabla_\theta g(v_2, \theta^*(v_2))||$$
$$= ||\nabla_2 [g(v_1, \theta^*(v_2)) - g(v_2, \theta^*(v_2))]||$$
$$\leq ||\nabla_{[1,2]} [g(v_1, \theta^*(v_2)) - g(v_2, \theta^*(v_2))]||$$
$$\leq L||v_1 - v_2||,$$

where $\nabla_{[1,2]}$ denotes taking the derivative on both first and second variables. We thus conclude that

$$||\theta^*(v_2) - \theta^*(v_1)|| \leq \frac{2L}{\kappa}||v_1 - v_2||.$$

### C.5.4 Proof of Lemma 4

To prove the first property,
$$||\nabla_\theta q(v, \theta_1) - \nabla_\theta q(v, \theta_2)|| = ||\nabla_\theta g(v, \theta_1) - \nabla_\theta g(v, \theta_2)||$$
$$\leq L||\theta_1 - \theta_2||.$$

Also
$$||\nabla q(v_1, \theta_1) - \nabla q(v_2, \theta_2)|| = ||\nabla g(v_1, \theta_1) - \nabla g(v_2, \theta_2) - \nabla g(v_1, \theta^*(v_1)) + \nabla g(v_2, \theta^*(v_2))||$$
$$\leq ||\nabla g(v_1, \theta_1) - \nabla g(v_2, \theta_2)|| + ||\nabla_1 g(v_1, \theta^*(v_1)) - \nabla_1 g(v_2, \theta^*(v_2))||.$$

By Assumption 2 (Lipschitz continuity of $\nabla g$),
$$||\nabla_1 g(v_1, \theta^*(v_1)) - \nabla_1 g(v_2, \theta^*(v_2))|| \leq ||\nabla_{[1,2]} g(v_1, \theta^*(v_1)) - \nabla_{[1,2]} g(v_2, \theta^*(v_2))||$$
$$\leq L\sqrt{||\theta^*(v_1) - \theta^*(v_2)||^2 + ||v_1 - v_2||^2},$$
where $\nabla_{[1,2]}$ denotes taking the derivative on both first and second variable. Also By Lemma 3,
$$L\sqrt{||\theta^*(v_1) - \theta^*(v_2)||^2 + ||v_1 - v_2||^2} \leq L\sqrt{\frac{4L^2}{\kappa^2}||v_1 - v_2||^2 + ||v_1 - v_2||^2}$$
$$\leq L(\frac{2L}{\kappa} + 1)||v_1 - v_2||.$$

This gives that
$$||\nabla q(v_1, \theta_1) - \nabla q(v_2, \theta_2)|| \leq ||\nabla g(v_1, \theta_1) - \nabla g(v_2, \theta_2)|| + ||\nabla_1 g(v_1, \theta^*(v_1)) - \nabla_1 g(v_2, \theta^*(v_2))||$$
$$\leq L\sqrt{||v_1 - v_2||^2 + ||\theta_1 - \theta_2||^2} + ||\nabla_1 g(v_1, \theta^*(v_1)) - \nabla_1 g(v_2, \theta^*(v_2))||$$
$$\leq L\sqrt{||v_1 - v_2||^2 + ||\theta_1 - \theta_2||^2} + L(\frac{2L}{\kappa} + 1)||v_1 - v_2||$$
$$\leq L_q\sqrt{||v_1 - v_2||^2 + ||\theta_1 - \theta_2||^2},$$
where $L_q := 2L(L/\kappa + 1)$.

### C.5.5 Proof of Lemma 5

By Lemma 4, we have
$$q(v, \theta^{(t+1)}) - q(v, \theta^{(t)}) \leq -(\alpha - \frac{L\alpha^2}{2})||\nabla_\theta q(v, \theta^{(t)})||^2.$$
By Assumption 1, we have
$$||\nabla_\theta q(v, \theta^{(t)})||^2 = ||\nabla_2 g(v, \theta^{(t)})||^2 \geq \kappa(g(v, \theta^{(t)}) - g(v, \theta^*(v)) = \kappa q(v, \theta^{(t)}).$$
Plug-in, we have
$$q(v, \theta^{(t+1)}) \leq (1 - (\alpha - \frac{L\alpha^2}{2})\kappa)q(v, \theta^{(t)}).$$
Recursively apply this inequality, we have
$$q(v, \theta^{(t)}) \leq (1 - (\alpha - \frac{L\alpha^2}{2})\kappa)^t q(v, \theta).$$
Let $b_1(r, L, \kappa) = \log(1 - (\alpha - L\alpha^2/2)\kappa)$, we have the desired result.

### C.5.6 Proof of Lemma 6

Notice that $||\nabla q(v, \theta)|| \leq ||\nabla g(v, \theta)|| + ||\nabla g(v, \theta^*(v))|| \leq 2M$. $||\hat{\nabla} q(v, \theta)|| \leq ||\nabla_v g(v, \theta)|| + ||\nabla_1 g(v, \theta^{(T)})|| + ||\nabla_\theta g(v, \theta)|| \leq 3M$. When $||\hat{\nabla} q|| = 0$, $||\delta^*|| = ||\nabla f|| \leq M$. When $||\hat{\nabla} q|| > 0$,
$$||\delta^*|| = ||[\eta||\hat{\nabla} q||^2 - \left\langle \nabla f, \hat{\nabla} q \right\rangle]_+ / ||\hat{\nabla} q||^2 \hat{\nabla} q + \nabla f||$$
$$\leq \eta||\hat{\nabla} q|| + 2||\nabla f|| \leq (2 + \eta)M.$$
This concludes that $||\delta^*|| \leq (2 + \eta)M$.

### C.5.7 Proof of Lemma 7

In the case that $\left\langle \nabla f, \hat{\nabla} q \right\rangle < \eta ||\hat{\nabla} q||^2$, $\lambda^* ||\hat{\nabla} q||^2 = \eta ||\hat{\nabla} q||^2 - \left\langle \nabla f, \hat{\nabla} q \right\rangle$. In the other case, $\lambda^* ||\hat{\nabla} q||^2 = 0$. Thus in all cases,

$$\lambda^* ||\hat{\nabla} q||^2 \leq \eta ||\hat{\nabla} q||^2 + ||\nabla f|| \, ||\hat{\nabla} q||$$
$$\leq \eta ||\hat{\nabla} q||^2 + M ||\hat{\nabla} q||.$$

### C.5.8 Proof of Lemma 8

Notice that since $\nabla q(v, \theta^*(v)) = 0$, we have

$$||\nabla q(v, \theta)|| = ||\nabla q(v, \theta) - \nabla q(v, \theta^*(v))|| \leq L_q ||\theta - \theta^*(v)|| \leq 2\kappa^{-1/2} L_q q^{1/2}(v, \theta),$$

where the first inequality is by Lemma 4 and the second inequality is by Lemma 1.

## D  Proof of the Result in Section 4.3

We use $b$ with some subscript to denote some general $O(1)$ constant and refer reader to section E for their detailed value.

For notation simplicity, given $v$ and $\theta$, $\theta^{(T)}$ denotes the results of $T$ steps of gradient of $g(v, \cdot)$ w.r.t. $\theta$ starting from $\theta$ using step size $\alpha$ (similar to the definition in (8)). And note that $\hat{\nabla} q(v, \theta) = \nabla g(v, \theta) - \left[ \nabla_1^\top g(v, \theta^{(T)}), \mathbf{0}^\top \right]^\top$, where $\mathbf{0}$ denotes a zero vector with the same dimension as $\theta$. We refer readers to the beginning of Appendix C for a discussion on the design of this extra notation and how it relates to the notation we used in Section 3. For simplicity, we omit the superscript $\diamond$ in $q^\diamond$ and simply use $q$ to denote $q^\diamond$ in the proof.

We start with the following two Lemmas.

**Lemma 13.** *Under Assumption 4 and assume $\alpha \leq 1/L$, for any $v, \theta$, $g(v, \theta) - g(v, \theta^\diamond(v, \theta)) \geq \frac{\kappa}{4} ||\theta - \theta^\diamond(v, \theta)||^2$.*

*Proof.* It is easy to show that

$$g(v, \theta^{(t+1)}) \leq g(v, \theta^{(t)}) - (\alpha - \frac{L\alpha^2}{2}) ||\nabla_\theta g(v, \theta^{(t)})||^2 \leq g(v, \theta^{(t)}).$$

We thus have $g(v, \theta^\diamond(v, \theta)) \leq g(v, \theta)$. The result of the proof follows the proof of Theorem 2 in Karimi et al. [24]. $\square$

**Lemma 14.** *Under Assumption 2 and 4, $||\theta^\diamond(v_2, \theta) - \theta^\diamond(v_1, \theta)|| \leq \frac{4L}{\kappa} ||v_1 - v_2||$ for any $v_1, v_2$.*

*Proof.* Notice that $\nabla q(v_2, \theta^\diamond(v_2, \theta)) = 0$, we have

$$||\nabla q(v_1, \theta^\diamond(v_2, \theta)) - \nabla q(v_2, \theta^\diamond(v_2, \theta))|| = ||\nabla q(v_1, \theta^\diamond(v_2, \theta))||.$$

By Assumption 4, we have $||\nabla q(v_1, \theta^\diamond(v_2, \theta))|| \geq \sqrt{\kappa(g(v_1, \theta^\diamond(v_2, \theta)) - g(v_1, \theta^\diamond(v_1, \theta)))}$. And by Lemma 13, we have

$$g(v_1, \theta^\diamond(v_2, \theta)) - g(v_1, \theta^\diamond(v_1, \theta)) \geq \frac{\kappa}{4} ||\theta^\diamond(v_2, \theta) - \theta^\diamond(v_1, \theta)||^2.$$

Combing all bounds gives that

$$2L ||v_1 - v_2|| \geq ||\nabla q(v_1, \theta^\diamond(v_2, \theta)) - \nabla q(v_2, \theta^\diamond(v_2, \theta))|| = ||\nabla q(v_1, \theta^\diamond(v_2, \theta))|| \geq \frac{\kappa}{2} ||\theta^\diamond(v_2, \theta) - \theta^\diamond(v_1, \theta)||.$$

This implies that $||\theta^\diamond(v_2, \theta) - \theta^\diamond(v_1, \theta)|| \leq \frac{4L}{\kappa} ||v_1 - v_2||$. $\square$

Now we proceed to give the proof of Theorem 2.

Note that

$$
\begin{aligned}
q(v_{k+1}, \theta_{k+1}) - q(v_k, \theta_k) &= [g(v_{k+1}, \theta_{k+1}) - g(v_{k+1}, \theta^\diamond(v_{k+1}, \theta_{k+1}))] - [g(v_k, \theta_k) - g(v_k, \theta^\diamond(v_k, \theta_k))] \\
&= [g(v_{k+1}, \theta_{k+1}) - g(v_{k+1}, \theta^\diamond(v_{k+1}, \theta_k))] - [g(v_k, \theta_k) - g(v_k, \theta^\diamond(v_k, \theta_k))] \\
&\quad + [g(v_{k+1}, \theta^\diamond(v_{k+1}, \theta_k)) - g(v_{k+1}, \theta^\diamond(v_{k+1}, \theta_{k+1}))] \\
&= [g(v_{k+1}, \theta_{k+1}) - g(v_k, \theta_k)] - [g(v_{k+1}, \theta^\diamond(v_{k+1}, \theta_k)) - g(v_k, \theta^\diamond(v_k, \theta_k))] \\
&\quad + [g(v_{k+1}, \theta^\diamond(v_{k+1}, \theta_k)) - g(v_{k+1}, \theta^\diamond(v_{k+1}, \theta_{k+1}))].
\end{aligned}
$$

Note that

$$
g(v_{k+1}, \theta_{k+1}) - g(v_k, \theta_k) \leq -\xi \left\langle \nabla g(v_k, \theta_k), \delta^*(v_k, \theta_k) \right\rangle + \frac{L\xi^2}{2} ||\delta^*(v_k, \theta_k)||^2
$$

$$
\begin{aligned}
-[g(v_{k+1}, \theta^\diamond(v_{k+1}, \theta_k)) - g(v_k, \theta^\diamond(v_k, \theta_k))] &\leq \left\langle \nabla_{[1,2]} g(v_k, \theta^\diamond(v_k, \theta_k)), [v_{k+1}, \theta^\diamond(v_{k+1}, \theta_k)] - [v_k, \theta^\diamond(v_k, \theta_k)] \right\rangle \\
&\quad + \frac{L}{2} ||[v_{k+1}, \theta^\diamond(v_{k+1}, \theta_k)] - [v_k, \theta^\diamond(v_k, \theta_k)]||^2.
\end{aligned}
$$

Notice that as $\nabla_2 g(v_k, \theta^\diamond(v_k, \theta_k)) = 0$,

$$
\left\langle \nabla_{[1,2]} g(v_k, \theta^\diamond(v_k, \theta_k)), [v_{k+1}, \theta^\diamond(v_{k+1}, \theta_k)] - [v_k, \theta^\diamond(v_k, \theta_k)] \right\rangle = \xi \left\langle \nabla_{[1,2]} g(v_k, \theta^\diamond(v_k, \theta_k)), \delta^*(v_k, \theta_k) \right\rangle.
$$

Also using Lemma 14, we have

$$
||\theta^\diamond(v_{k+1}, \theta_k) - \theta^\diamond(v_k, \theta_k)|| \leq \frac{4L}{\kappa} ||v_{k+1} - v_k||.
$$

This implies that

$$
||[v_{k+1}, \theta^\diamond(v_{k+1}, \theta_k)] - [v_k, \theta^\diamond(v_k, \theta_k)]||^2 \leq (\frac{16L^2}{\kappa^2} + 1) ||v_{k+1} - v_k||^2 \leq (\frac{16L^2}{\kappa^2} + 1) \xi^2 ||\delta^*(v_k, \theta_k)||^2.
$$

We thus have

$$
q(v_{k+1}, \theta_{k+1}) - q(v_k, \theta_k) \leq -\xi \left\langle \nabla q(v_k, \theta_k), \delta^*(v_k, \theta_k) \right\rangle + L_q \xi^2 ||\delta^*(v_k, \theta_k)||^2/2 + \chi_k,
$$

where we define $L_q = (\frac{16L^2}{\kappa^2} + 2)$ and $\chi_k = [g(v_{k+1}, \theta^\diamond(v_{k+1}, \theta_k)) - g(v_{k+1}, \theta^\diamond(v_{k+1}, \theta_{k+1}))]$.
Using the same argument in the proof of Lemma 10 and Lemma 11, we have

$$
\begin{aligned}
& q(v_{k+1}, \theta_{k+1}) - q(v_k, \theta_k) \\
&\leq -\xi\eta ||\nabla q(v_k, \theta_k)||^2 + 12\xi\eta L_q^2 \kappa^{-1} \exp(-b_1 T) q(v_k, \theta_k) \\
&\quad + 2\xi b_2 L \kappa^{-1/2} \exp(-b_1 T/2) q^{1/2}(v_k, \theta_k) + L_q \xi^2 b_2^2/2 + \chi_k \\
&\leq -\xi\eta ||\nabla q(v_k, \theta_k)||^2 + 12\xi\eta L_q^2 \kappa^{-2} \exp(-b_1 T) ||\nabla q(v_k, \theta_k)||^2 \\
&\quad + 2\xi b_2 L \kappa^{-1} \exp(-b_1 T/2) ||\nabla q(v_k, \theta_k)|| + L_q \xi^2 b_2^2/2 + \chi_k.
\end{aligned}
$$

Here the second inequality is by Assumption 4. Choosing $T$ such that $T \geq b_7(\eta, \alpha, \kappa, L)$ where

$$
b_7(\eta, \alpha, \kappa, L) = \left\lceil -b_1^{-1} \log(\frac{\kappa^2}{48\eta L_q^2}) \right\rceil,
$$

we have

$$
q(v_{k+1}, \theta_{k+1}) - q(v_k, \theta_k) \leq -\frac{3}{4} \xi\eta ||\nabla q(v_k, \theta_k)||^2 + 2\xi b_2 L \kappa^{-1} \exp(-b_1 T/2) ||\nabla q(v_k, \theta_k)|| + L_q \xi^2 b_2^2/2 + \chi_k.
$$

Using Young's inequality, given any $x > 0$,

$$
\exp(-b_1 T/2) ||\nabla q(v_k, \theta_k)|| \leq x \exp(-b_1 T) + \frac{1}{x} ||\nabla q(v_k, \theta_k)||^2.
$$

Choosing $x = \frac{4Lb_2}{\eta\kappa}$, we have

$$
q(v_{k+1}, \theta_{k+1}) - q(v_k, \theta_k) \leq -\frac{1}{4} \xi\eta ||\nabla q(v_k, \theta_k)||^2 + \Delta + \chi_k,
$$

where we denote $\Delta = \xi \frac{8L^2 b_2^2}{\eta \kappa^2} \exp(-b_1 T) + \frac{1}{2} L_q \xi^2 b_2^2$. This gives that

$$\frac{1}{4}\xi\eta \sum_{k=0}^{K} ||\nabla q(v_k, \theta_k)||^2 \leq q(v_0, \theta_0) - q(v_K, \theta_K) + K\Delta + \sum_{k=0}^{K-1} \chi_k.$$

Using the same argument in the proof of Lemma 11,

$$||\hat{\nabla} q(v, \theta)|| \leq (2L\kappa^{-1} + 1)||\nabla q(v, \theta)||.$$

We hence have

$$\sum_{k=0}^{K-1} ||\hat{\nabla} q(v_k, \theta_k)||^2 \leq (2L\kappa^{-1} + 1)^2 \sum_{k=0}^{K-1} ||\nabla q(v_k, \theta_k)||^2$$

$$\leq \frac{4(2L\kappa^{-1} + 1)^2}{\xi\eta}(q(v_0, \theta_0) - q(v_K, \theta_K) + K\Delta + \sum_{k=0}^{K-1} \chi_k).$$

Similar to the proof of Lemma 12,

$$\sum_{k=0}^{K-1} ||\delta^*(v_k, \theta_k)||^2 \leq \frac{2(f(v_0, \theta_0) - f(v_K, \theta_K))}{\xi} + 2\eta^2 \sum_{k=0}^{K-1} ||\hat{\nabla} q(v_k, \theta_k)||^2 + 2\eta M\sqrt{K}\sqrt{\sum_{k=0}^{K-1} ||\hat{\nabla} q(v_k, \theta_k)||^2}.$$

Using $\sqrt{x + y} \leq \sqrt{x} + \sqrt{y}$, we have

$$2\eta^2 \sum_{k=0}^{K-1} ||\hat{\nabla} q(v_k, \theta_k)||^2 \leq \frac{8\eta(2L\kappa^{-1} + 1)^2}{\xi}(q(v_0, \theta_0) - q(v_K, \theta_K) + K\Delta + \sum_{k=0}^{K-1} \chi_k)$$

$$2\eta M\sqrt{K}\sqrt{\sum_{k=0}^{K-1} ||\hat{\nabla} q(v_k, \theta_k)||^2} \leq \sqrt{K}\frac{4\eta^{1/2} M(2L\kappa^{-1} + 1)}{\xi^{1/2}}(\sqrt{q(v_0, \theta_0) - q(v_K, \theta_K)} + K^{1/2}\Delta^{1/2} + \sqrt{\left[\sum_{k=0}^{K-1} \chi_k\right]_+}).$$

Also notice that by Assumption 4,

$$\sum_{k=0}^{K-1} q(v_k, \theta_k) \leq \sum_{k=0}^{K-1} \frac{\xi}{\kappa}||\nabla q(v_k, \theta_k)||^2$$

$$\leq \frac{4}{\eta\kappa\xi}(q(v_0, \theta_0) - q(v_K, \theta_K) + K\Delta + \sum_{k=0}^{K-1} \chi_k)$$

We hence have

$$\sum_{k=0}^{K-1} (||\delta^*(v_k, \theta_k)||^2 + q(v_k, \theta_k)) = O\left(\frac{1}{\xi} + \frac{K\Delta}{\xi} + \frac{K^{1/2}}{\xi^{1/2}} + \frac{K\Delta^{1/2}}{\xi^{1/2}} + K^{1/2}\left(\left[\sum_{k=0}^{K-1} \chi_k\right]_+\right)^{1/2}\right)$$

$$= O\left(\frac{1}{\xi} + K\exp(-b_1 T/2) + K\xi^{1/2} + \frac{K^{1/2}}{\xi^{1/2}} + \left(K\left[\sum_{k=0}^{K-1} \chi_k\right]_+\right)^{1/2}\right).$$

Using the same argument as the proof of Theorem 1, when $T \geq \lceil -b_1^{-1} \log(\frac{1}{16}\kappa^2 L^{-2})\rceil$,

$$\mathcal{K}^\diamond(v, \theta) \leq 2||\delta^*(v, \theta)||^2 + q(v, \theta) + 8L^2 \exp(-b_1 T)\kappa^{-2}(\eta + 2)^2 b_2^2.$$

This implies that

$$\min_k \mathcal{K}^\diamond(v_k, \theta_k) \leq \frac{1}{K}\sum_{k=0}^{K-1} [2||\delta^*(v, \theta)||^2 + q(v, \theta)] + 8L^2 \exp(-b_1 T)\kappa^{-2}(\eta + 2)^2 b_2^2$$

$$= O\left(\frac{1}{\xi K} + \exp(-b_1 T/2) + \xi^{1/2} + \frac{1}{\xi^{1/2} K^{1/2}} + \left(\left[\frac{1}{K}\sum_{k=0}^{K-1} \chi_k\right]_+\right)^{1/2}\right).$$

Now we proceed to bound $\frac{1}{K}\sum_{k=0}^{K-1}\chi_k$. Notice that

$$\chi_k = g(v_{k+1}, \theta^\diamond(v_{k+1}, \theta_k)) - g(v_{k+1}, \theta^\diamond(v_{k+1}, \theta_{k+1}))$$
$$= g(v_{k+1}, \theta^\diamond(v_{k+1}, \theta_k)) - g(v_k, \theta^\diamond(v_k, \theta_k)) + g(v_k, \theta^\diamond(v_k, \theta_k)) - g(v_{k+1}, \theta^\diamond(v_{k+1}, \theta_{k+1})).$$

Notice that using Assumption 2 and Lemma 14

$$g(v_{k+1}, \theta^\diamond(v_{k+1}, \theta_k)) - g(v_k, \theta^\diamond(v_k, \theta_k)) \le L||[v_{k+1}, \theta^\diamond(v_{k+1}, \theta_k)] - [v_k, \theta^\diamond(v_k, \theta_k)]||$$
$$\le L(||v_{k+1} - v_k|| + ||\theta^\diamond(v_{k+1}, \theta_k) - \theta^\diamond(v_k, \theta_k)||)$$
$$\le (L + \frac{4L}{\kappa})||v_{k+1} - v_k||$$
$$\le (L + \frac{4L}{\kappa})\xi||\delta^*(v_k, \theta_k)||.$$

Note that using the same procedure as the proof of Lemma 6, $||\delta^*(v_k, \theta_k)|| \le b_2$. We thus conclude that

$$\sum_{k=0}^{K-1} \chi_k \le \sum_{k=0}^{K-1} g(v_k, \theta^\diamond(v_k, \theta_k)) - g(v_{k+1}, \theta^\diamond(v_{k+1}, \theta_{k+1}))$$
$$+ (L + \frac{4L}{\kappa})\xi \sum_{k=0}^{K-1} ||\delta^*(v_k, \theta_k)||$$
$$\le \sum_{k=0}^{K-1} g(v_k, \theta^\diamond(v_k, \theta_k)) - g(v_{k+1}, \theta^\diamond(v_{k+1}, \theta_{k+1})) + (L + \frac{4L}{\kappa})b_2\xi K$$
$$= g(v_0, \theta^\diamond(v_0, \theta_0)) - g(v_K, \theta^\diamond(v_K, \theta_K)) + (L + \frac{4L}{\kappa})b_2\xi K.$$

We thus have $\frac{1}{K}\sum_{k=0}^{K-1}\chi_k = O(\frac{1}{K} + \xi)$.

# E    List of absolute constants used in the proofs

Here we summarize the absolute constant used in the proofs.

$$b_1(\alpha, L, \kappa) = \log(1 - (\alpha - L\alpha^2/2)\kappa)$$
$$b_2(M, \eta) = (3 + \eta)M$$
$$b_3(\eta, \alpha, \kappa, L) = \left\lceil -b_1^{-1}\log(\frac{\eta\kappa}{64\eta L_q^2}) \right\rceil$$
$$b_4(\eta, \kappa, \xi) = -\log(1 - \frac{\xi}{4}\eta\kappa)$$
$$b_5(L_q, \eta, \kappa) = \frac{16L_q^2}{\eta\kappa^2}(2L\kappa^{-1} + 1)^2$$
$$b_6(\kappa, L) = (2L\kappa^{-1} + 1)^2$$
$$b_7(\eta, \alpha, \kappa, L) = \left\lceil -b_1^{-1}\log(\frac{\kappa^2}{48\eta L_q^2}) \right\rceil$$