# OpenReview forum: "BOME! Bilevel Optimization Made Easy: A Simple First-Order Approach"
_NeurIPS.cc/2022/Conference — NeurIPS 2022 Accept_

### Official Review · Reviewer_DoMn · 2022-07-10

**Rating:** 6
**Confidence:** 4
**Soundness:** 3 good
**Presentation:** 4 excellent
**Contribution:** 3 good

**Summary:**

This paper proposes a first-order bilevel algorithm which updates only based on first-order gradient information. The author provides convergence analysis for such algorithm and conducts different experiments to demonstrates its superior performances across different settings.

**Questions:**

1. It will be better to have a discussion about the comparison between convergence results of proposed algorithm and other benchmarks (e.g. AID, ITD, VRBO).

2. In terms of experiments, it will be better to include several stochastic algorithms, e.g. (SUSTAIN [24], StocBiO [21])
 for comparison to check if BOME ranks top among these algorithms.

3. The continual learning experiments (Table 1) do not show the superior performances of BOME compared with BVFSM. However, in Section 6.2, the author still claims that "BOME outerforms BVFSM across all experiments".

4. In Line 290, the author states that FT=ACC+NBT. However, the results in Table 1 do not support such states. Any possible explanation?



**Limitations:**

Non negative societal impact.

**Strengths And Weaknesses:**

In terms of strengths, it is interesting to use value function approach to avoid Hessian computation in bilevel optimization. Such design helps to reduce the computational cost. Proposed algorithm has been demonstrated in various settings. In general, this paper is well written and presented.

In terms of weakness, the convergence result is not standard (not in gradient norm form) which make it hard to compare such result with other benchmarks. In experiments, all algorithms have adopted GD to update while some algorithms (e.g. VRBO) is designed for update stochastically. It will be better to involve stochastic algorithms for comparison. In continual learning experiment, CTN (+BOME) does outperform CTN (+BVFSM).

---

> ### Author Response · Authors · 2022-08-02
> **Authors' response to Reviewer DoMn**
>
> We thank reviewer DoMn for the comments and feedback!
>
> **Comparing convergence results**
>
> We thank the reviewer for the suggestions! The other more classic approaches frame their optimization of BO as a standard gradient descent problem using the idea of hyper-gradient. However, such results assume LLS structure and require Hessian computation at each iteration, so their computational complexity at each iteration is much higher.
>
> In comparison, we consider a more general, possibly nonconvex lower level problem, and we do not require any second-order information. Hence, we think these earlier methods are not comparable to the LLS and Hessian-free algorithm proposed in our paper. We think our 'non-standard' result should be viewed as a technical novelty rather than a weakness, that is, we establish a novel analysis for a new type of algorithm.
>
> Finally, we want to point out that non-asymptotic results of different BO algorithms are still largely open and, to the best of our knowledge, we believe that we are the first to establish a non-asymptotic rate of a purely first-order BO algorithm under reasonable assumptions.
> These things said, we will add more discussion to compare our theoretical results and the existing ones.
>
> **Compare with stochastic algorithms**
>
> We thank the reviewer for the suggestion. We have included StocBiO in our experiment. We want to also emphasize that VRBO we compared within our initial submission is also a stochastic algorithm. Regarding stocBiO, we use their official implementation from https://github.com/JunjieYang97/stocBiO, but apply it to our problems. Notice that though stocBiO also applies to MNIST dataset, the training loss they plot (last plot from the GitHub repo) is difficult to interpret, since the test loss on average is above 2.3 (e.g. ln(10)), which is similar to the loss from random prediction. The only difference between our problem and theirs is that we initialize the random matrix in a more reasonable way so that for the initial choice, the training/test loss is around ln(10) = 2.3. We performed a grid search over the hyperparameters of stocBiO around their suggested value but found that stocBiO, though having low per-iteration times, decreases the loss more slowly than other methods.
>
> **BOME outperforms BVFSM on continual learning**
>
> We thank the reviewer for asking. The goal of continual learning is to balance forgetting (backward transfer) and fast adaptation (forward transfer). Therefore, all CL methods naturally are searching on the Pareto frontier formed by the two objectives, and it is natural to see one having better NBT and the other having better FT. In general, the most important metric in CL is the average ACC after learning the final task. From this perspective, BOME indeed achieves a better point on the Pareto frontier than BVFSM.
>
>
> **On FT=ACC+NBT**
>
> We thank the reviewer for catching this. The problem is that when calculating FT and ACC, we are averaging over the $t$ tasks seen so far. But for NBT, since the $t$-th task is just learned and does not have any forgetting, we are averaging over $(t-1)$ tasks instead. This fact, together with small rounding errors, leads to inconsistency. We updated Table 1 accordingly. Again, as we mentioned in the response to the previous question, the most important metric here is the ACC, which balances both NBT and FT.

---

> > ### Comment · Reviewer_DoMn · 2022-08-08
> > **Thank you for the response**
> >
> > Thank you for the response. Most of my questions have been solved. However, I have a little concern about the experimental results and the authors do not show SUSTAIN result. Hence, I remain my score.

---

> > > ### Author Response · Authors · 2022-08-08
> > > **Further response to Reviewer DoMn**
> > >
> > > We thank the reviewer again for providing those positive comments and suggestions!
> > >
> > > Regarding the experimental results, we wonder which part of the experiment concerns the reviewer? If it is the SUSTAIN result, the problem is that we have not found any open source implementation of the SUSTAIN algorithm online. If the reviewer can point us to any reference implementation, we are happy to include the comparison against SUSTAIN in our experiment. On the other hand, we want to point out that we have included both VRBO and stocBiO into our experiment, both of which are stochastic methods and have been shown to outperform SUSTAIN (please refer to Fig. 1 from https://arxiv.org/pdf/2106.04692.pdf).
> > >
> > > From the algorithm perspective, we believe SUSTAIN and BOME are addressing different challenges in the bilevel optimization. Namely, SUSTAIN focuses on designing a stochastic momentum-assisted gradient estimator for both the upper and lower level updates, while BOME considers how to solve the bilevel optimization problem with purely first-order information.
> > >
> > > From the theory perspective, SUSTAIN assumes the lower level subproblem is strongly convex, while we do not assume convexity for the lower level problem. We want to convince the reviewer that both the problem setting we consider (e.g. purely first-order solver) and the theoretical analysis we present (e.g. result when the lower level subproblem is non-convex) are relatively significant contributions for the bilevel optimization community. In addition, we will release the code for reproducing our result, with the hope to benefit future research.

---

> > > > ### Comment · Reviewer_DoMn · 2022-08-08
> > > > **Thanks for the further response**
> > > >
> > > > Based on my knowledge, SUSTAIN is the same as MRBO (https://arxiv.org/pdf/2106.04692.pdf) but with batch size 1. Both them are momentum-based algorithms. Hence, it will be good to make comparison between proposed algorithms and these two.
> > > >
> > > > MRBO code is available online (https://github.com/JunjieYang97/MRVRBO). It will be very useful if you can release your code then.
> > > >
> > > > If current algorithms show better performance than MRBO & SUSTAIN, I will increase my score to 6.

---

> > > > > ### Author Response · Authors · 2022-08-09
> > > > > **Followup for Reviewer DoMn**
> > > > >
> > > > > We sincerely thank the reviewer for the reference and suggestion. Due to the limited time remaining, we could not systematically run hyperparameter search for SUSTAIN and MRVRBO on our problems using our implementation. Moreover, simply incorporating MRBO's implementation into our codebase results in failure to learn.
> > > > >
> > > > > Therefore, we decided to implement BOME on SUSTAIN and MRVRBO's codebase, **the result is included in the supplementary material**. Note that we directly cloned the repo from the referenced github website and used the official implementation of MRBO. From the figure, one can see that BOME largely outperforms both methods. Specifically, we observe that SUSTAIN (due to its small batch size), fails to learn well.

---

> > > > > > ### Comment · Reviewer_DoMn · 2022-08-09
> > > > > > **Thanks for the response and experiments**
> > > > > >
> > > > > > Thanks for the response. I have increased the score to 6.

---

### Official Review · Reviewer_wdMq · 2022-07-11

**Rating:** 4
**Confidence:** 5
**Soundness:** 4 excellent
**Presentation:** 3 good
**Contribution:** 2 fair

**Summary:**

This paper proposes a fast bi-level optimization (BO) algorithm, that only involves the calculation of the first-order gradient. The proposed method is built on the value-function approach, and integrated the dynamic barrier gradient descent algorithm into the new BO algorithm, which helps solve the BO problem more efficiently. The proposed method is a fully first-order method and the authors also establish the non-asymptotic convergence analysis for the proposed method. Experiments on toy examples as well as real deep learning tasks are conducted to demonstrate the effectiveness of the proposed method.

**Questions:**

Please address the comments and questions above in the "weakness" column.

**Limitations:**

The authors have adequately addressed the limitations and potential negative societal impact of their work.

**Strengths And Weaknesses:**

Strengths:

1. Theoretical Analysis: The convergence analysis for the proposed method is indeed one contribution to the bi-level community. The discussion on the condition of the lower-level objective $g$ is also remarkable. I understand when $g$ has multiple local minima, the analysis could become very tricky. Therefore, the introduction of the attraction points is very impressive.

2. Presentation: This paper is well-organized, straight forward and in general very easy to follow. I have no difficulty in grasping the main idea as well as the technical details of the proposed method.

Weakness:

1. Novelty: In general, I think the novelty of this paper is less significant and incremental. Firstly, the value function based bi-level optimization algorithm is well studied and known to not require the computation of the Hessian matrix. Secondly, as the authors mentioned in the paper for several times, the core algorithm used in the proposed method "Dynamic Barrier Gradient Descent" is an existing approach. The authors have yet to convince me that the integration of a proposed method is significant enough. The authors should make the difference between the proposed method and the existing method more distinctive, or discuss the essential difference between these two methods.

2. Experiments: The authors claim that their method is good at large-scale practical deep learning tasks. However, the current experiments all seem to be far away from "large-scale". Therefore, the authors are suggested to design some larger scale experiments to verify their argument. The current results all look good to me.

3. The authors are also suggested to do an additional experiments on the few-shot meta-learning tasks, since this task is very widely accepted as one experiment to examine the feasibility of a BO method.

4. Regarding the hyper-parameters, the author claim that "BOME is robust to the choice of $\eta$, T and $\alpha$ as varying them results in almost identical performance". I think this is counter-intuitive, since more inner steps T generally indicates a better lower-level solution, closer to the stationary point, which could lead to possible better results. Further, why do you choose $\eta=0.5$? Is there any logics behind this?

---

> ### Author Response · Authors · 2022-08-02
> **Authors' response to Reviewer wdMq**
>
> We thank reviewer wdMq for the constructive comments!
>
> **On novelty of BOME**
>
> We appreciate the reviewer for raising this concern and agree on the need for a more detailed discussion on the difference between BOME and Gong et al.. Below are the main points.
>
> First of all, from the problem formulation perspective, one can think of the problems that Gong et al. consider are bilevel optimizations (BO) without the outer variable (i.e. $v$ in the paper). By contrast, BOME aims to solve the general BO problem, therefore enjoys a wider range of applications in practice.
>
> Algorithmically speaking, extending Gong et al. to the general BO problem is not straightforward, especially when the lower level problem is non-convex. Developing a fully first-order algorithm for general BO requires introducing the stop-gradient operation in a mathematically correct way - we believe BOME’s integration of those techniques is also novel. There are several details about BOME which seem simple but are subtle and critical. For example, when using gradient descent to compute $\theta_k^T$, simply changing its starting point (line 109) from $\theta_k$ to $\theta_{k-1}^T$ (which is the design choice of many existing value-based such as [1]) will break both the theoretical result and the empirical performance when the lower level problem is nonconvex. Overall, we believe it is not necessary for every paper to propose an entirely new framework. We think our contribution to developing a purely first-order BO algorithm is also valuable.
>
> From the theoretical analysis perspective, as also mentioned by the reviewer, establishing the non-asymptotic convergence rate is far more challenging than the analysis given by Gong et al.. Indeed, non-asymptotic results have not yet been obtained for many BO algorithms. Even for the classic hypergradient-based approach (such results are established only recently [2]). We believe that we are the first to establish a non-asymptotic rate for a purely first-order BO algorithm under reasonable assumptions, e.g. the lower level problem can be both convex or non-convex.
>
> Empirically, we conduct a comprehensive analysis of BOME on illustrative toy examples and demonstrate that it achieves state-of-art performance on various important BO benchmark problems, including problems that the dynamic barrier gradient descent is not able to solve.
>
>
> **On large-scale experiment**
>
> Thanks for asking. The experiments on continual learning (CL) are already large-scale: the underlying backbone we use is modified from a standard ResNet-18, which is much larger than most neural networks used in previous work. In addition, the CL objective is quite similar to the meta-learning objective, since the goal is to optimize one part of the model to support the fast adaptation of the other part of the model.
>
> **On hyper-parameters $T$**
>
> In theory, larger $T$ should always help but as suggested by our theory result, we have different convergence rates in terms of $T$ and the outer iteration $K$. We, in general, expect an exponential convergence for $T$ while a polynomial rate of $K$, suggesting that we do not require a large number of inner steps.
>
> On the other hand, when the inner variable $\theta$ is close to some attraction points, and the outer variable changes only slightly when the objectives are smooth, the inner variable will still be close to the new attraction point, and hence $T$ might not need to be large.
>
> **On hyper-parameters $\eta$**
>
> In general, there are no restrictions on choosing $\eta$. Empirically, we find $\eta \in [0,1]$ is reasonable and we simply pick $\eta = 0.5$ and find it is robust across experiments.
>
> **Reference**
>
> [1] A Value-Function-based Interior-point Method for Non-convex Bi-level Optimization.
>
> [2] Bilevel optimization: Convergence analysis and enhanced design.

---

> ### Author Response · Authors · 2022-08-08
> **Any follow-up questions?**
>
> Dear reviewer wdMq, thanks again for your time and effort for reviewing. We see that you’ve acknowledged our rebuttal and we were wondering if there are any of the points you raised up that haven’t been properly address? Thanks very much!

---

### Official Review · Reviewer_RuUT · 2022-07-11

**Rating:** 7
**Confidence:** 2
**Soundness:** 3 good
**Presentation:** 3 good
**Contribution:** 3 good

**Summary:**

The authors present a first-order smooth bilevel optimization algorithm that avoids implicit differentiation, instead using a value function to reduce the problem to a single level constrained optimization problem. The single level problem is solved by a so-called dynamic barrier gradient descent algorithm, which notably does not require the lower level problem to have a unique solution. Convergence guarantees are proven for the algorithm and numerical experiments are reported on several different examples, confirming the theoretical results and suggesting that the algorithm performs favorably in practice.

**Questions:**

In line 113, where it's explained that theta T is treated as constant despite depending on v, how is this justified? Even if you had a global minimizer theta star instead of the approximation, it would still depend on v, so it's unclear if this works.

In line 44, what is meant by powerful BO method?

In line 82 and line 107, if the lower level problem is nonconvex then what is meant by theta star?

In line 138, should it be CRCQ instead of CRCO?

**Limitations:**

I think the paper has done a good job explaining the limitations of its contributions, which are primarily theoretical analyses, by clearly stating the assumptions needed to apply the theorems.

**Strengths And Weaknesses:**

I found the paper to be generally well written and clear, save for a few small typos that otherwise don't affect the arguments, and I appreciated that the assumptions on the functions and problem settings were clearly stated, adding to the quality of the paper.

While the idea of using a value function in bilevel optimization to reduce the problem to a single level optimization problem isn't original per se, the algorithm and analysis here seem original and significant to the best of my knowledge, particularly the convergence rate for the case where the lower level solution is not unique. The analysis given doesn't require necessarily the uniqueness of the minimizer for the lower level problem, which makes the algorithm more broadly applicable than many other bilevel optimization algorithms, adding to the significance of the paper.

There were a lot of different problems considered in the applications section and the results were fairly convincing, although I think that in the examples which admit a unique minimizer for the lower level problem it would be interesting to compare to AmIGO as well.

---

> ### Author Response · Authors · 2022-08-02
> **Authors' response to Reviewer RuUT**
>
> We thank reviewer RuUT for the positive feedback and comments!
>
> **Compare with AmIGO**
>
> We thank the reviewer for the suggestion and have added the reference to AmIGO in the updated paper. Since AmIGO is a recent work and we have not yet found any open-source implementation of it (the official repository states the code is coming soon), we will wait for its release to compare it against BOME. We point out, however, that AmIGO is an enhanced method based on implicit differentiation and not a first-order method.
>
> **On $\theta^T$ is treated as constant despite depending on $v$**
>
> $\theta^T$ depends on v. Here we mean that when taking the gradient $\nabla \hat{q}$, we will view $\theta^T$ as a constant rather than a function of v. To be more concrete, if we view $\theta^T$ as a function of v, we obtain
>
> $$\nabla_{v}g(v,\theta_{k}^{(T)})=\nabla_{1}g(v,\theta_{k}^{(T)})+\nabla_{2}g(v,\theta_{k}^{(T)})\nabla_{v}\theta_{k}^{(T)},$$
>
> where $\nabla_i$ means taking the gradient of the $i$-th variable.
> Viewing  $\theta^T$ as a constant, we have
>
> $$\nabla_{v}g(v,\theta_{k}^{(T)})=\nabla_{1}g(v,\theta_{k}^{(T)}).$$
>
> In this case, we require only first-order information. Thanks for raising this point.
>
> **Regarding BOME is a power method**
>
> The proposed method BOME is a general first-order approach and works well in many practical examples. In these senses, we believe that it is 'powerful'.
>
> **$\theta^{*}$ when lower level problem is nonconvex**
>
> Thanks for pointing it out! $\theta^{*}(v)$ means the optimal value of $\theta$ given $v$, and it might be intractable to compute when the lower level problem is nonconvex. When considering/analyzing the *practical* algorithm for the nonconvex lower level problem, we seek instead the local attraction point $\theta^\diamond$. Essentially, we change the lower level problem from one of global optimization to one of local optimization.
>
> **Typos**
>
> Thanks for pointing out the typo of CRCQ. We have fixed it in the updated manuscript.

---

> > ### Comment · Reviewer_RuUT · 2022-08-07
> > **Treating theta T as constant**
> >
> > Your response regarding theta T being treated as constant did not address my concerns - I understand well the difference in the calculation but I don't understand the justification. It does depend on v so how can you justify ignoring this dependence? It is also somehow a big deal, since if you do the calculation with the dependence on v then you have to involve what the paper calls second-order information, defeating the motivation.
> >
> > Besides this, my concerns have all been adequately addressed.

---

> > > ### Author Response · Authors · 2022-08-08
> > > **Follow up**
> > >
> > > Thank you for clarifying your question!
> > >
> > > One way to justify is to view it as a plug-in estimator. We know that
> > > $$
> > > \nabla{v_k}q(v_k,\theta_k) = \nabla_{v_k}g(v_k,\theta_k) - \nabla_{v_k}g^{*}(v_k)
> > > $$
> > >
> > > And
> > >
> > > $$
> > > \nabla_{v_k} g^{\*}(v_k)  = \nabla_{1} g(v_k, {\theta}^{\*}(v_k) ) + \nabla_{v_k} \theta^{\*}(v_k) \nabla_{2}g(v_k, \theta^{*}(v_k)).
> > > $$
> > >
> > > Note that since $\nabla_{2}g(v_k, \theta^{\*}(v_k)) = 0$ by the definition of the optimum $\theta^{*}(v_k)$, we thus have
> > >
> > > $$
> > > \nabla_{v_{k}}q(v_{k},\theta_{k})=\nabla_{v_{k}}g(v_{k},\theta_{k})-\nabla_{1}g(v_{k},\theta^{*}(v_{k})).
> > > $$
> > >
> > > In practical algorithm, since we do not know $\theta^{\*}(v_{k})$, we estimate $\nabla_{v_{k}}q(v_{k},\theta_{k})$, intuitively, by pluging-in an estimator of $\theta^{\*}(v_{k})$ (which is $\theta_{k}^{(T)}$ obtained by performing $T$-steps' gradient starting from $\theta_{k}$.). This results in the following estimator of the gradient
> > >
> > > $$
> > > \hat{\nabla_{v_{k}}} q(v_{k},\theta_{k}) =\nabla_{v_{k}}g(v_{k},\theta_{k})-\nabla_{1}g(v_{k},\theta_{k}^{(T)}).
> > > $$
> > >
> > > Equivalently, this can be viewed as the gradient of function $\hat{q}(v,\theta)$ where $\theta_{k}^{(T)}$ is treated as a constant.
> > >
> > > Our theory proves that using such approach, we are able to converge to a local optimum of the problem based on the theory of constrained qualification. Lemma 10 gives the characterization on how such approach is able to decrease the loss of the inner problem --- when $T$ is reasonable large, $\theta_{k}^{(T)}$ eventually well approximates $\theta^{\*}(v_{k})$ and gives a good gradient.

---

### Official Review · Reviewer_67cV · 2022-07-19

**Rating:** 6
**Confidence:** 2
**Soundness:** 3 good
**Presentation:** 3 good
**Contribution:** 3 good

**Summary:**

The paper studies the bilevel optimization with unique inner solution, proposed the algorithm that utilizes the dynamic barrier gradient methods, theoretically captures the convergence rate of the proposed algorithm, and implemented plenty of numerical simulations to ensure the validity of the proposed algorithm.

**Questions:**

Since we do not have the access to the second order information of $\nabla^2 g(v, \theta)$, do we implicitly approximately computed the inverse of Hessian?

**Limitations:**

It would be better if the authors provide more intuition on how the convergence rate is derived.

**Strengths And Weaknesses:**

The paper is convincing that it provides various numerical simulations that verifies the validity of the algorithm, and it also provides the convergence rate, not just asymptotic guarantee.

---

> ### Author Response · Authors · 2022-08-02
> **Authors' response to Reviewer 67cV**
>
> We thank reviewer 67cV for the review and comments!
>
> **Question on second order information**
>
> Thanks for asking! The major advantage of BOME is that it does not even approximate the inverse of Hessian, and is, therefore, a *fully* first-order method. In contrast, methods like conjugate gradient require iterative computation to bypass the computation of Hessian, while other methods like iterative differentiation (ITD) require unrolling the computation graph and therefore involve multiple Jacobian vector multiplications.
>
> **Intuition on how the convergence rate is derived**
>
> We tried our best to present the main analysis in a concise way. The organization of the proof section in the Appendix already provides a sketch of how we approach the rate: with several helper Lemmas being extracted, the proof of the main theorems is within one page. The key idea is to study how the outer and inner objectives evolve during the training, with the focus on how the gradient deviates from the oracle gradient when the value-based constraint is only approximately satisfied.

---

> > ### Comment · Reviewer_67cV · 2022-08-07
> > **Thank you for the response**
> >
> > Your answer solves my questions, but I remain the score.

---

### Meta-Review · Area_Chair_zR5Y · 2022-08-26

**Recommendation:** Accept
**Confidence:** Less certain

**Metareview:**

There is general consensus among the reviewers that this paper is a valuable contribution to the bilevel optimization literature.

- The value function formulation is still relatively unexplored in bilevel optimization (althought not completely new). Having a new paper developing this direction will be a nice addition to the literature.
- The paper seems well written and sound.
- The experiments, though they don't really assess the scalability of the approach, are illustrative and diverse.

We therefore recommend acceptance.

To the authors: please take into account the reviewer comments in the camera-ready paper. Please be careful of not overselling the contributions with superlatives like "powerful method".

**Award:**

No

---

### Decision · Program_Chairs · 2022-09-14

Accept